# On Class Distributions Induced by Nearest Neighbor Graphs for Node Classification of Tabular Data

**Federico Errica**
NEC Laboratories Europe
Heidelberg, Germany

## Abstract

Researchers have used nearest neighbor graphs to transform classical machine learning problems on tabular data into node classification tasks to solve with graph representation learning methods. Such artificial structures often reflect the homophily assumption, believed to be a key factor in the performances of deep graph networks. In light of recent results demystifying these beliefs, we introduce a theoretical framework to understand the benefits of Nearest Neighbor (NN) graphs when a graph structure is missing. We formally analyze the Cross-Class Neighborhood Similarity (CCNS), used to empirically evaluate the usefulness of structures, in the context of nearest neighbor graphs. Moreover, we study the class separability induced by deep graph networks on a $k$-NN graph. Motivated by the theory, our quantitative experiments demonstrate that, under full supervision, employing a $k$-NN graph offers no benefits compared to a structure-agnostic baseline. Qualitative analyses suggest that our framework is good at estimating the CCNS and hint at $k$-NN graphs never being useful for such classification tasks under full supervision, thus advocating for the study of alternative graph construction techniques in combination with deep graph networks.

## 1 Introduction

The pursuit of understanding real-world phenomena has often led researchers to model the system of interest as a set of interdependent constituents, which influence each other in complex ways. In disciplines such as chemistry, physics, and network science, graphs are a convenient and well-studied mathematical object to represent such interacting entities and their attributes. In machine learning, the term "graph representation learning" refers to methods that can automatically leverage graph-structured data to solve tasks such as entity (or node), link, and whole-graph predictions [12, 32, 78, 73, 4].

Most of these methods assume that the relational information, that is the connections between entities, naturally emerges from the domain of the problem and is thus known. There is also broad consensus that connected entities typically share characteristics, behavioral patterns, or affiliation, something known as the homophily assumption [45]. This is possibly why, when the structure is *not* available, researchers have tried to artificially build Nearest Neighbor (NN) graphs from tabular data, by connecting entities based on some attribute similarity criterion, with applications in healthcare [44, 65], fake news and spam detection [39, 7], biology [77, 42], and document classification [56] to name a few. From an information-theoretic perspective, the creation of such graphs does not add new information as it depends on the available data; that said, what makes their use plausible is that the graph construction is a form of feature engineering that often encodes the homophily assumption. Combined with the inductive bias of Deep Graph Networks (DGNs) [59, 46], this strategy aims at improving the generalization performances on tabular data compared to structure-agnostic baselines, for example, a Multi-Layer Perceptron (MLP).

37th Conference on Neural Information Processing Systems (NeurIPS 2023).

Indeed, using a $k$-NN graph [21] has recently improved the node classification performances *under the scarcity of training labels* [24, 23]. This is also known as the semi-supervised setting, where one can access the features of all nodes but the class labels are available for a handful of those. A potential explanation for these results is that, by incorporating neighboring values into each entity's representation, the neighborhood aggregation performed by DGNs acts as a regularization strategy that prevents the classifier from overfitting the attributes of the few labeled nodes, similar to input jittering [57, 62, 6]. However, it is still unclear what happens *when one has access to all training labels* (hereinafter the fully-supervised setting), namely if these graph-building strategies grant a statistically significant advantage in generalization compared to a structure-agnostic baseline. In this respect, proper comparisons against such baselines are often lacking or unclear in previous works, an issue that has also been reported in recent papers about the reproducibility of node and graph classification experiments [61, 22, 54].

In addition, it was recently shown [43] that homophily is not required to achieve good classification performances in node classification tasks; rather, what truly matters is how different the neighborhood class label distributions of nodes of separate classes are. This resulted in the definition of the *empirical* Cross-Class Neighborhood Similarity (CCNS) [43, 13], an object that estimates such similarities based on the *available* connectivity structure. Yet, whether or not artificially built graphs can be useful for the task at hand has mainly remained an empirical question, and more theoretical conditions for which this happens are still not understood.

In this paper, we introduce a theoretical framework to approach this question, and we provide two analyses of independent interest. Inspired by the CCNS, Section 3.1 studies the neighboring class label distribution of nearest neighbor graphs. Then Section 3.2 deals with the distribution of entity embeddings induced by DGNs on a $k$-NN graph, and we use it to quantify class separability in both the input and the embedding spaces. Overall, our theoretical results suggest that building a $k$-NN graph is not a good idea. To validate the theory with empirical evidence, we robustly compare (Sections 4 and 5) four baselines across 11 tabular datasets to check that the $k$-NN graph construction does not give statistically significant advantages in the fully-supervised setting. In addition, we reverse-engineer the theory to learn data distributions that would make a $k$-NN graph useful in practice when combined with DGNs. From the empirical results, we conjecture that this is never the case. Therefore, we hope to raise awareness in the graph machine learning community about the need for alternative (either NN-based or not) graph construction techniques.

In summary, our contributions are: *i)* under some assumptions, we can estimate the CCNS for nearest neighbor graphs and provide its first lower bound; *ii)* we study how applying a simple DGN to an artificial $k$-NN graph affects the class separability of the input data; *iii)* we carry out a robust comparison between structure-agnostic and structure-aware baselines on a set of 11 datasets that is in agreement with our theoretical results; *iv)* qualitative analyses based on the theory further suggest that using the $k$-NN graph might not be advisable.

## 2   Related Work

The early days of graph representation learning date back to the end of the previous century, when researchers proposed backpropagation through structures to process directed acyclic graphs [63, 25]. These ideas laid the foundations for the adaptive processing of cyclic graphs by the recurrent Graph Neural Network [59] and the feedforward Neural Network for Graphs [46], which today's DGNs build upon. Both methods iteratively compute embeddings of the graphs' entities (also called nodes) via a local and iterative message passing mechanism [27] that propagates the information through the graph. In recent years, many neural and probabilistic DGNs have emerged [50, 36, 31, 69, 74, 3] bridging ideas from different fields of machine learning; we set up our analysis in the context of these message-passing architectures. Even more recently, transformer models have begun to appear in graph-related tasks as well [19, 37, 75]. Akin to kernel methods for graphs [71, 38], this class of methods mainly relies on feature engineering to extract rich information from the input graph, and some perform very well at molecular tasks [41]. However, the architecture of transformers is not intrinsically more expressive than DGNs, and their effectiveness depends on the specific encodings used [48]. Therefore, gaining a better understanding of the inductive bias of DGNs remains a compelling research question.

The construction of NN graphs found recent application in predicting the mortality of patients, by connecting them according to specific attributes of the electronic health records [44, 65]. In addition, it was used in natural language processing to connect messages and news with similar contents to tackle spam and fake news detection, respectively [39, 7]. In both cases, the authors computed similarity based on some embedding representation of the text, and the terms' frequency in a document was the graph-building criterion for a generic document classification task [56]. Finally, $k$-NN graphs have also been built based on chest computerized tomography similarity for early diagnoses of COVID-19 [77, 42] or to understand if debts will be repaid in time [29].

Most of the theoretical works on DGNs deal with the problems of over-smoothing [40, 15, 14, 11] and over-squashing [1, 66, 28] of learned representations, as well as the discriminative power of such models [47, 74, 26, 10]. In this context, researchers mostly agreed that DGNs based on message passing perform favorably for homophilic graphs and not so much for heterophilic ones [79]. However, recent works suggest a different perspective; the generalization performances depend more on the neighborhood distributions of nodes belonging to different classes [43] and on a good choice of the model's weights [67]. The Cross-Class Neighborhood Similarity (CCNS) Ma et al. [43] stands out as an effective (but purely empirical) strategy to understand whether a graph structure is useful or not for a node classification task. In our work, we take inspiration from the CCNS to study the behavior of the neighborhood class label distributions around nodes and compute the first approximation and lower bound of the CCNS for NN graphs without the need to explicitly build them.

Structure learning and graph rewiring are also related but orthogonal topics. Rather than pre-computing a fixed structure, these approaches discover latent dependencies between samples [24, 72] and can enrich the original graph structure when this is available [17, 76, 66, 34]. They have been applied in contexts of scarce supervision, where a $k$-NN graph proved to be an effective baseline when combined with DGNs [23]. At the same time, the combinatorial nature of graphs makes it difficult and expensive to explore the space of all possible structures, making the a priori construction of the graph a sensible and efficient alternative.

## 3 Methodology

We begin by introducing notions and assumptions that will be used throughout the paper. Our starting point is a classification task over a set of classes $\mathcal{C}$, where each sample $u$ is associated with a vector of $D$ attributes $\boldsymbol{x}_u \in \mathbb{R}^D$ and a target class label $y_u \in \mathcal{C}$.

A graph $g$ of size $N$ is a quadruple $(\mathcal{V}, \mathcal{E}, \mathcal{X}, \mathcal{Y})$, where $\mathcal{V} = \{1, \dots, N\}$ represents the set of nodes and $\mathcal{E}$ is the set of *directed* edges $(u, v)$ from node $u$ to node $v$. The symbol $\mathcal{X} = \{X_u, \forall u \in \mathcal{V}\}$ defines the set of random variables (*r.v.*) with realizations $\boldsymbol{x}_u \in \mathbb{R}^D, u \in \mathcal{V}$. The same definition applies to the set of target variables $\mathcal{Y} = \{Y_u, \forall u \in \mathcal{V}\}$ and their realizations $y_u \in \mathcal{C}, u \in \mathcal{V}$. When we talk about the subset of nodes with target label $c \in \mathcal{C}$, we use the symbol $\mathcal{V}_c$; also, we define the *neighborhood* of node $u$ as $\mathcal{N}_u = \{v \in \mathcal{V} | (v, u) \in \mathcal{E}\}$.

A Gaussian (or normal) univariate distribution with mean and standard deviation $\mu, \sigma \in \mathbb{R}, \sigma > 0$ is represented as $\mathcal{N}(\cdot; \mu, \sigma^2)$, using $\boldsymbol{\mu} \in \mathbb{R}^D, \boldsymbol{\Sigma} \in \mathbb{R}^{D \times D}$ for the multivariate case. The probability density function (*p.d.f.*) of a univariate normal *r.v.* parametrized by $\mu, \sigma$ is denoted by $\phi(\cdot)$ together with its cumulative density function (*c.d.f.*) $F(w) = \boldsymbol{\Phi}(\frac{w-\mu}{\sigma}) = \frac{1}{2}(1 + \text{erf}(\frac{w-\mu}{\sigma\sqrt{2}}))$, where erf is the error function. Subscripts will denote quantities related to a specific random variable.

We want to transform the initial task into a node classification problem, where different *i.i.d.* samples become the nodes of a single graph, and the edges are computed by some nearest neighbor algorithm. We assume (*assumption* 1) the true data distribution $p(X = \boldsymbol{x})$ of each sample (thus abstracting from $u$) is defined by the graphical model of Figure 1 (left). This model reflects the common assumption that observables of a given class share the same distribution [30, 70], and we choose this distribution to be a mixture of Gaussian distributions. Such a choice is not restrictive since, in principle, with enough mixture components it is possible to approximate almost any continuous density [9]. Note that it is also possible to sample discrete attribute values by rounding the samples of a properly parametrized Gaussian mixture. Formally, we consider $|\mathcal{C}|$ latent classes modeled by a categorical *r.v.* $C$ with prior distribution $p(C = c)$ and $|\mathcal{M}|$ mixtures for each class modeled by $M \sim p(M = m | C = c), m \in \mathcal{M}, c \in \mathcal{C}$, such that $p(\boldsymbol{x}|c) = \sum_{m=1}^{\mathcal{M}} p(\boldsymbol{x}|m, c)p(m|c)$. In Section 3.1, we further require (*assumption* 2) that the

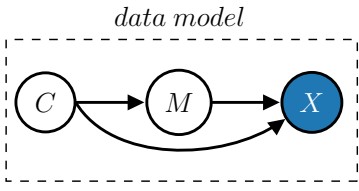

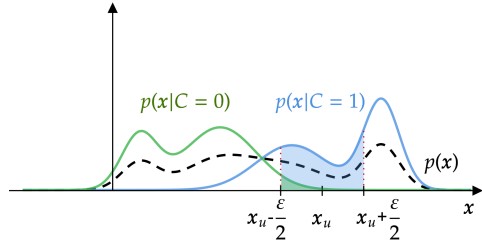

Figure 1: *Left:* we represent assumption 1 on the data distribution through a graphical model, i.e., a hierarchical mixture of distributions, where observed variables are shaded and latent ones are white circles. *Right:* a visual example of a data distribution generated by the graphical model, with an intuitive visualization of the class posterior mass around a point $\boldsymbol{x}_u$.

$D$ observables are conditionally independent when the class $c$ and the mixture $m$ are known, i.e., $p(X = \boldsymbol{x}) = \sum_{c=1}^{|\mathcal{C}|} p(c) \sum_{m=1}^{|\mathcal{M}|} p(m|c) \prod_{f=1}^{D} p(x_f|m, c)$. We refer the reader to Section 3.3 for a discussion about this assumption. Figure 1 (right) depicts an example of one such data distribution for $D = 1$. Hereinafter we shall have $p(X_f = x_f|m, c) = \mathcal{N}(x_f; \mu_{cmf}, \sigma_{cmf}^2)$ for the individual attributes and $p(X = \boldsymbol{x}|m, c) = \mathcal{N}(\boldsymbol{x}; \boldsymbol{\mu}_{mc}, \boldsymbol{\Sigma}_{mc})$ for the multivariate case. We will also use diagonal covariance matrices denoted as $\boldsymbol{\Lambda}_{mc} = diag(\sigma_{mc1}^2, \ldots, \sigma_{mcD}^2)$. Finally, akin to [43], we assume that the neighbors of node $v$ are *i.i.d.* (*assumption* 3), meaning their attributes and class distributions only depend on the properties of node $v$; such an assumption is especially reasonable when one considers how nearest neighbor structures are typically built, by picking a set of neighbors based only on their distance to the target node $v$.

To talk about the surroundings of a point in space we will use the notion of hypercubes (or D-cubes). A hypercube of dimension $D$ centered at point $\boldsymbol{x} \in \mathbb{R}^D$ of side length $\varepsilon > 0$ is the set of points given by the Cartesian product $H_\varepsilon(\boldsymbol{x}) = [x_1 - \frac{\varepsilon}{2}, x_1 + \frac{\varepsilon}{2}] \times \cdots \times [x_D - \frac{\varepsilon}{2}, x_D + \frac{\varepsilon}{2}]$.

### 3.1 Analytical Computation of the CCNS for Nearest Neighbor Graphs

The CCNS [43] computes how similar the neighborhoods of two distinct nodes are in terms of class label distribution, and it provides an aggregated result over pairs of target classes. Intuitively, if nodes belonging to distinct classes happen to have similar neighboring class label distributions, then it will be unlikely that a classifier will correctly discriminate between these two nodes after a message passing operation because the nodes' embeddings will look very similar. On the other hand, nodes of different classes with very different neighboring class label distributions will be easier to separate. This intuition implies that nodes of different classes typically have different attribute values.

**Definition 3.1** (Cross Class Neighborhood Similarity, extended from [43]). Given a graph $g$, the cross-class neighborhood similarity between classes $c, c' \in \mathcal{C}$ is given by

$$s(c, c') = \mathbb{E}_{p(\boldsymbol{x}|c)p(\boldsymbol{x}'|c')}[\Omega(q_c(\boldsymbol{x}), q_{c'}(\boldsymbol{x}'))] \tag{1}$$

where $\Omega$ computes a (dis)similarity score between vectors and the function $q_c : \mathbb{R}^D \to \mathbb{R}^{|\mathcal{C}|}$ (resp $q_{c'}$) computes the probability vector that a node of class $c$ (resp. $c'$) with attributes $\boldsymbol{x}$ (resp. $\boldsymbol{x}'$) has a neighbor of class $c''$, for every $c'' \in \mathcal{C}$.

The definition of $q_c$ and $q_{c'}$ is the key ingredient of Equation 1. In the following, we are going to show that it is possible to analytically compute these quantities when we assume an NN structure. For every norm-induced metric, which is also convex, the lower bound of Equation 1 follows from Jensen's inequality and from the linearity of expectation:

$$s(c, c') \geq || \mathbb{E}_{p(\boldsymbol{x}|c)p(\boldsymbol{x}'|c')}[q_c(\boldsymbol{x}) - q_{c'}(\boldsymbol{x}')] || = || \mathbb{E}_{p(\boldsymbol{x}|c)}[q_c(\boldsymbol{x})] - \mathbb{E}_{p(\boldsymbol{x}'|c')}[q_{c'}(\boldsymbol{x}')] ||. \tag{2}$$

This bound assigns non-zero values to the *inter-class* neighborhood similarity, but as we shall see one can resort to Monte Carlo approximations of Equation 1 to estimate both the *inter* and *intra-class* scores. From now on, we will use the Euclidean distance as our $\Omega$.

We first study the class label distribution in the surroundings of some node $u$. In the example of Figure 1 (right), we consider a binary classification problem with $|\mathcal{M}| = 2$ and depict the conditional distributions $p(\boldsymbol{x}_u | C = 0)$, $p(\boldsymbol{x}_u | C = 1)$ with green and blue curves, respectively. A dashed black line, instead, represents $p(\boldsymbol{x})$ assuming a non-informative class prior. If the neighbors of $u$ belong to the hypercube $H_\varepsilon(\boldsymbol{x}_u)$ for some $\varepsilon$, then the probability that a neighbor will belong to class $c$ depends on how much class-specific probability mass, that is the shaded green and blue areas, there is in the hypercube. Since the blue area is larger than the green one, finding a neighbor of class 1 is more likely to happen. Formally, we define the (unnormalized) probability of a neighbor belonging to class $c$ in a given hypercube as the weighted posterior mass of $C$ contained in that hypercube.

**Definition 3.2** (Posterior Mass $M_{\boldsymbol{x}}(c, \varepsilon)$ Around Point $\boldsymbol{x}$). Given a hypercube $H_\varepsilon(\boldsymbol{x})$ centered at point $\boldsymbol{x} \in \mathbb{R}^D$, and a class $c \in \mathcal{C}$, the posterior mass $M_{\boldsymbol{x}}(c, \varepsilon)$ is the unnormalized probability that a point in the hypercube has class $c$:

$$M_{\boldsymbol{x}}(c, \varepsilon) = \mathbb{E}_{\boldsymbol{w} \in H_\varepsilon(\boldsymbol{x})}[p(c|\boldsymbol{w})] = \int_{\boldsymbol{w} \in H_\varepsilon(\boldsymbol{x})} p(c|\boldsymbol{w})p(\boldsymbol{w})d\boldsymbol{w} = p(c) \int_{\boldsymbol{w} \in H_\varepsilon(\boldsymbol{x})} p(\boldsymbol{w}|c)d\boldsymbol{w}, \quad (3)$$

where the last equality follows from Bayes' theorem.

The following proposition shows how to compute $M_{\boldsymbol{x}}(c, \varepsilon)$ analytically. The interested reader can find all proofs of our paper in Section A.3.

**Proposition 3.3.** *Under assumptions 1,2, $M_{\boldsymbol{x}}(c, \varepsilon)$ has the following analytical form*

$$p(c) \sum_{m=1}^{|\mathcal{M}|} p(m|c) \prod_{f=1}^{D} \left( F_{Z_{cmf}}\left(x_f + \frac{\varepsilon}{2}\right) - F_{Z_{cmf}}\left(x_f - \frac{\varepsilon}{2}\right) \right), \quad (4)$$

*where $Z_{cmf}$ is a r.v. with Gaussian distribution $\mathcal{N}(\cdot; \mu_{cmf}, \sigma_{cmf}^2)$.*

To reason about an entire class rather than individual samples, therefore being able to compute the two quantities on the right-hand side of Equation 2, we extend the previous definition by taking into account all samples of class $c' \in \mathcal{C}$. Thus, we seek to compute the average unnormalized probability that a sample belongs to class $c$ in the hypercubes centered around samples of class $c'$.

**Definition 3.4** (Expected Class $c$ Posterior Mass $M_{c'}(c, \varepsilon)$ for Samples of Class $c'$). Given a hypercube length $\varepsilon$ and a class $c' \in \mathcal{C}$, the unnormalized probability that a sample of class $c'$ has another sample of class $c$ lying in its hypercube is defined as

$$M_{c'}(c, \varepsilon) \stackrel{\text{def}}{=} \mathbb{E}_{\boldsymbol{x} \sim p(\boldsymbol{x}|c')}[M_{\boldsymbol{x}}(c, \varepsilon)]. \quad (5)$$

It is also possible to compute $M_{c'}(c, \varepsilon)$ in closed form, which we show in the next theorem.

**Theorem 3.5.** *Under assumptions 1,2, $M_{c'}(c, \varepsilon)$ has the following analytical form*

$$p(c) \sum_{\substack{m=1 \\ m'=1}}^{|\mathcal{M}|} p(m'|c')p(m|c) \prod_{f=1}^{D} \left( \boldsymbol{\Phi}\left( \frac{\mu_{c'm'f} + \frac{\varepsilon}{2} - \mu_{cmf}}{\sqrt{\sigma_{cmf}^2 + \sigma_{c'm'f}^2}} \right) - \boldsymbol{\Phi}\left( \frac{\mu_{c'm'f} - \frac{\varepsilon}{2} - \mu_{cmf}}{\sqrt{\sigma_{cmf}^2 + \sigma_{c'm'f}^2}} \right) \right). \quad (6)$$

Thanks to Theorem 3.5, we know how much class-$c$ posterior probability mass we have, on average, around samples of class $c'$. To get a proper class $c'$-specific distribution over neighboring class labels, we apply a normalization step using the fact that $M_{c'}(c, \varepsilon) \geq 0 \ \forall c \in \mathcal{C}$.

**Definition 3.6** ($\varepsilon$-Neighboring Class Distribution). Given a class $c'$ and an $\varepsilon > 0$, the neighboring class distribution around samples of class $c'$ is modeled as

$$p_{c'}(c, \varepsilon) = \frac{M_{c'}(c, \varepsilon)}{\sum_{i=1}^{|\mathcal{C}|} M_{c'}(i, \varepsilon)}. \quad (7)$$

This distribution formalizes the notion that, in a neighborhood around points of class $c'$, the probability that points belong to class $c$ does not necessarily match the true prior distribution $p(C = c)$. However, this becomes false when we consider an infinitely-large hyper-cube, as we show in the next result.

**Proposition 3.7.** *Let the first $D$ derivatives of $\sum_{i=1}^{|\mathcal{C}|} M_{c'}(i)$ be different from 0 in an open interval $I$ around $\varepsilon = 0$ and not tending to infinity for $\varepsilon \to 0$. Under assumptions 1,2, Equation 7 has limits*

$$\lim_{\varepsilon \to 0} p_{c'}(c, \varepsilon) = \frac{p(c) \sum_{m,m'}^{|\mathcal{M}|} p(m|c) p(m'|c') \prod_{f=1}^{D} \phi_{Z_{cmm'f}}(0)}{\sum_{i}^{|\mathcal{C}|} p(i) \sum_{m,m'}^{|\mathcal{M}|} p(m|i) p(m'|c') \prod_{f=1}^{D} \phi_{Z_{imm'f}}(0)}, \qquad \lim_{\varepsilon \to \infty} p_{c'}(c, \varepsilon) = p(c)$$

(8)

*where $Z_{imm'f}$ has distribution $\mathcal{N}(\cdot; -a_{imm'f}, b_{imm'f}^2)$,*

$a_{imm'f} = 2(\mu_{c'm'f} - \mu_{imf})$ *and* $b_{imm'f} = 2\sqrt{\sigma_{imf}^2 + \sigma_{c'm'f}^2}$.

The choice of $\varepsilon$, which implicitly encodes our definition of "nearest" neighbor, plays a crucial role in determining the distribution of a neighbor's class label. When the hypercube is too big, the probability that a neighbor has class $c$ matches the true prior $p(c)$ regardless of the class $c'$, that is we make a crude assumption about the neighbor's class distribution of a sample. If we instead consider a smaller hypercube, then we observe a less trivial behavior and the probability $p_{c'}(c, \varepsilon)$ is directly proportional to the distances between the means $\boldsymbol{\mu}_{c'm'}$ and $\boldsymbol{\mu}_{cm}$, as one would intuitively expect for simple examples.

To summarize, we can use $p_{c'}(c, \varepsilon)$ as an approximation for $\mathbb{E}_{p(\boldsymbol{x}|c')}[q'_c(\boldsymbol{x})]$ of Equation 2; similarly, we can use a normalized version of $M_c(\boldsymbol{x})$ in place of $q_c(\boldsymbol{x})$ to estimate Equation 1 via Monte Carlo sampling without the need of building a NN graph. For space reasons, in Appendix A.5, we qualitatively discuss the results for a 1-dimensional instance of a data distribution, and we investigate the quality of the CCNS approximation while varying $k$.

## 3.2 Class Separability Induced by DGNs on a $k$-NN Graph

This section formally investigates the properties of the embedding space created by DGNs under the $k$-NN graph. Our goal is to understand whether using such a graph can improve the separability between samples belonging to different classes or not. Provided that our assumptions hold, both conclusions would be interesting: if the $k$-NN graph helps, then we know the conditions for that to happen; if that is not the case, we have identified and formalized the need for new graph construction mechanisms.

Akin to previous works [40, 16, 5, 43], we consider a linear 1-layer DGN with the following neighborhood aggregation scheme that computes node embeddings $\boldsymbol{h}_u \in \mathbb{R}^D \; \forall u \in \mathcal{V}$:

$$\boldsymbol{h}_u = \frac{1}{k} \sum_{v \in \mathcal{N}_u} \boldsymbol{x}_v. \tag{9}$$

The node embedding of sample $u$ is then fed into a standard machine learning classifier, e.g., an MLP. As done in previous works, we assume that a linear (learnable) transformation $\boldsymbol{W} \in \mathbb{R}^{D \times D}$ of the input $\boldsymbol{x}_v$, often used in DGN models such as the Neural Network for Graphs [46] and the Graph Convolutional Network [36], is absorbed by the subsequent classifier.

Mimicking the behavior of the $k$-NN algorithm, which connects *similar* entities together, we model the attribute distribution of a neighbor $v \in \mathcal{N}_u$ as a normal distribution $\mathcal{N}(\boldsymbol{x}_v; \boldsymbol{x}_u, diag(\sigma^2, \ldots, \sigma^2))$, where $\sigma^2$ is a hyper-parameter that ensures it is highly unlikely to sample neighbors outside of $H_\varepsilon(\boldsymbol{x}_u)$; from now on we use the symbol $\sigma_\varepsilon$ to make this connection clear. Under assumption 3, neighbors' sampling is independently repeated $k$ times and the attributes are averaged together. Therefore, we use statistical properties of normal distributions (please refer to Lemmas A.2 and A.1) to compute the resulting node $u$'s embedding distribution:

$$p(\boldsymbol{h}_u | \boldsymbol{x}_u) = \mathcal{N}(\boldsymbol{h}_u; \boldsymbol{x}_u, \boldsymbol{\Lambda}_\varepsilon), \quad \boldsymbol{\Lambda}_\varepsilon = diag(\sigma_\varepsilon^2/k, \ldots, \sigma_\varepsilon^2/k). \tag{10}$$

Therefore, the more neighbors a node has (i.e., higher $k$) the more skewed the resulting distribution is around $\boldsymbol{x}_u$ (i.e., lower variance). This is a fairly reasonable assumption if we think of an infinitely large dataset; informally, each of the $k$ neighbors will be much more likely to lie near $\boldsymbol{x}_u$ than near the surface of the hypercube, hence $\boldsymbol{h}_u$ will be close to $\boldsymbol{x}_u$ with high probability.

To understand how Equation 9 affects the separability of samples belonging to different classes, we compute a divergence score between the distributions $p(\boldsymbol{h}|c)$ and $p(\boldsymbol{h}|c')$ [68, 51, 20, 8, 52]. When

this divergence is higher than that of the distributions $p(x|c)$ and $p(x|c')$, then the $k$-NN structure and the inductive bias of DGNs are helpful for our task. Below, we show that for two mixtures of Gaussians we can obtain the analytical form of their Squared Error Distance (SED), the simplest symmetric divergence defined as $SED(p, q) = \int (p(x) - q(x))^2 dx$. This provides a concrete strategy to understand, regardless of training, if it would make sense to build a $k$-NN graph for our problem.[1]

**Proposition 3.8.** *Under assumptions 1,3, let us assume that the entity embeddings of a 1-layer DGN applied on an artificially built $k$-NN graph follow the distribution of Equation 10. Then the Squared Error Distances $SED(p(\boldsymbol{h}|c), p(\boldsymbol{h}|c'))$ and $SED(p(\boldsymbol{x}|c), p(\boldsymbol{x}|c'))$ have analytical forms.*

In the interest of space, we have deferred the analytic formulas to the appendix.

As an immediate but important corollary, a $k$-NN graph improves the ability to distinguish samples of different classes $c, c'$ if it holds that $SED(p(\boldsymbol{h}|c), p(\boldsymbol{h}|c')) > SED(p(\boldsymbol{x}|c), p(\boldsymbol{x}|c'))$. Indeed, if class distributions diverge more in the embedding space, which has the same dimensionality as the input space, then they will be easier to separate by a universal approximator such as an MLP. We use this corollary in our experiments, by reverse-engineering the theory to find out "good" data models.

### 3.3 Limitations and Future Directions

It is extremely challenging to prove that a $k$-NN graph *never* helps to improve the classification performances using a DGN. The reason is that theoretical results of this kind rely, as is common [43], on assumptions of the true data generating distribution and on the specific assignments of parameters such as $\boldsymbol{\mu}$ and $\boldsymbol{\Sigma}$. The true data distribution is typically unknown and hard to find [68], but assumptions 1 and 3 remain quite reasonable due to the flexibility of mixture models and the nature of the nearest neighbors construction. In addition, assumption 2 has been widely used in the graphical models literature because it is particularly attractive from a theoretical perspective [64], despite it cannot cover all real-world cases; we note, however, that it plays no role in the analysis of Section 3.2. Finally, as done in the past [40, 15, 5, 43], we have ignored the non-linearity in the neighborhood aggregation of Equation 9. As we shall see in our empirical results, the intuitions we gained from the theoretical analysis seem to hold for non-linear models as well.
Our analysis has considered hyper-cubes in the interest of simplicity, but nearest neighbor graphs typically compute hyper-balls centered at a point. In high-dimensional spaces, the volume of hyper-cubes and hyper-balls can differ in non-trivial ways [18], so future work should improve on our results by replacing hyper-cubes with hyper-balls. Similarly, the results of Section 3.1 could be used to guide the definition of new graph construction strategies based on attribute similarity criteria, for instance by proposing a good $\varepsilon$ for the data at hand. We leave this idea to future investigations.

## 4 Experiments

We conduct quantitative and qualitative experiments to support our theoretical insights. For all experiments we used a server with 32 cores, 128 GBs of RAM, and 4 GPUs with 11 GBs of memory.

**Quantitative Experiments** Quantitatively speaking, we want to compare a structure-agnostic baseline, namely an MLP [58], against different graph machine learning models, such as a simple DGN (sDGN) that implements Equation 9 followed by an MLP classifier, the Graph Isomorphism Network (GIN) [74] and the Graph Convolutional Network (GCN) [36]. The goal is to show that in the fully-supervised setting using a $k$-NN graph does not offer any concrete benefit. We consider 11 datasets, eight of which were taken from the UCI repository [2], namely Abalone, Adult, Dry Bean, Electrical Grid Stability, Isolet, Musk v2, Occupancy Detection, Waveform Database Generator v2, as well as the citation networks Cora, Citeseer [60], and Pubmed [49]. For each dataset, we build a $k$-NN graph, where $k$ is a hyper-parameter, using the node attributes' similarity to find neighbors (discarding the original structure in the citation networks). We report some datasets statistics in Table 2, and the metric to optimize is the F1 score. Note that it is difficult to ascertain if our assumptions hold for these datasets because we have no access to the true data distribution; however, a robust comparison for this topic is lacking in the literature and can help us gather further insights on whether our ideas might also hold in a more general setting. For every dataset and model, we follow the rigorous and fair evaluation setup of [22]: we perform a 10-fold cross validation for risk assessment,

---

[1]The proof also makes a simple but meaningful connection to the over-smoothing problem.

Table 1: Node classification mean and standard deviation results for a structure-agnostic baseline (MLP) compared to graph machine learning methods applied to artificially built $k$-NN graphs. We performed model selection w.r.t. the F1 score, but we also include the accuracy for completeness.

| | MLP | | sDGN | | GIN | | GCN | |
|---|---|---|---|---|---|---|---|---|
| | F1 | ACC | F1 | ACC | F1 | ACC | F1 | ACC |
| Abalone | $\mathbf{0.55}_{\pm 0.02}$ | $55.5_{\pm 1.5}$ | $\mathbf{0.55}_{\pm 0.01}$ | $55.3_{\pm 1.2}$ | $0.54_{\pm 0.03}$ | $54.2_{\pm 1.7}$ | $0.54_{\pm 0.02}$ | $54.4_{\pm 1.7}$ |
| Adult | $0.70_{\pm 0.02}$ | $77.6_{\pm 1.7}$ | $\mathbf{0.72}_{\pm 0.02}$ | $77.8_{\pm 1.9}$ | $0.69_{\pm 0.02}$ | $76.2_{\pm 1.3}$ | $0.70_{\pm 0.02}$ | $78.6_{\pm 0.9}$ |
| DryBean | $\mathbf{0.78}_{\pm 0.04}$ | $77.4_{\pm 4.0}$ | $0.77_{\pm 0.06}$ | $77.3_{\pm 5.4}$ | $0.75_{\pm 0.03}$ | $74.4_{\pm 3.2}$ | $0.71_{\pm 0.07}$ | $72.6_{\pm 4.9}$ |
| eGrid | $\mathbf{0.95}_{\pm 0.01}$ | $95.6_{\pm 0.6}$ | $\mathbf{0.95}_{\pm 0.01}$ | $95.8_{\pm 0.4}$ | $0.93_{\pm 0.01}$ | $94.0_{\pm 0.6}$ | $0.82_{\pm 0.02}$ | $83.1_{\pm 1.4}$ |
| Isolet | $\mathbf{0.95}_{\pm 0.01}$ | $95.4_{\pm 0.5}$ | $\mathbf{0.95}_{\pm 0.00}$ | $95.4_{\pm 0.3}$ | $\mathbf{0.95}_{\pm 0.01}$ | $95.3_{\pm 0.7}$ | $0.92_{\pm 0.01}$ | $91.7_{\pm 1.4}$ |
| Musk | $\mathbf{0.99}_{\pm 0.01}$ | $99.3_{\pm 0.3}$ | $\mathbf{0.99}_{\pm 0.01}$ | $99.5_{\pm 0.3}$ | $0.94_{\pm 0.05}$ | $97.0_{\pm 2.2}$ | $0.94_{\pm 0.02}$ | $96.7_{\pm 0.9}$ |
| Occupancy | $\mathbf{0.99}_{\pm 0.01}$ | $98.9_{\pm 0.4}$ | $\mathbf{0.99}_{\pm 0.01}$ | $98.9_{\pm 0.5}$ | $0.98_{\pm 0.01}$ | $98.6_{\pm 0.6}$ | $0.90_{\pm 0.17}$ | $95.8_{\pm 6.0}$ |
| Waveform | $\mathbf{0.85}_{\pm 0.02}$ | $85.5_{\pm 1.7}$ | $\mathbf{0.85}_{\pm 0.02}$ | $85.3_{\pm 1.5}$ | $0.84_{\pm 0.02}$ | $84.5_{\pm 1.7}$ | $0.84_{\pm 0.03}$ | $84.0_{\pm 2.2}$ |
| Cora | $0.71_{\pm 0.04}$ | $74.3_{\pm 2.4}$ | $0.71_{\pm 0.03}$ | $74.1_{\pm 2.5}$ | $\mathbf{0.72}_{\pm 0.03}$ | $74.9_{\pm 2.4}$ | $0.70_{\pm 0.04}$ | $73.0_{\pm 2.6}$ |
| Citeseer | $0.69_{\pm 0.02}$ | $71.8_{\pm 1.9}$ | $0.69_{\pm 0.03}$ | $72.9_{\pm 2.7}$ | $0.69_{\pm 0.02}$ | $72.1_{\pm 2.2}$ | $\mathbf{0.71}_{\pm 0.03}$ | $74.4_{\pm 2.4}$ |
| Pubmed | $\mathbf{0.87}_{\pm 0.01}$ | $87.6_{\pm 0.7}$ | $\mathbf{0.87}_{\pm 0.01}$ | $87.6_{\pm 0.6}$ | $\mathbf{0.87}_{\pm 0.01}$ | $86.6_{\pm 0.9}$ | $0.83_{\pm 0.01}$ | $83.5_{\pm 0.9}$ |

with a hold-out model selection (90% training/10% validation) for each of the 10 external training folds. We report the hyper-parameters for the model selection phase in Table 3. For each of the 10 best models, we perform 3 final re-training runs and average their test performances on the corresponding external fold to mitigate bad initializations. The final score is the average of these 10 test results.

**Qualitative Experiments** We provide four qualitative experiments. First, we study how $p_c(c', \varepsilon)$ changes for varying values of $\varepsilon$ and different priors $p(c)$ under specific class-conditional data distributions. Second, we analyze the first limit of Proposition 3.7 under the same conditional data distributions as before ($D = 1$ hence assumptions are trivially satisfied). We translated one of the distributions by different values and computed the SED between them, together with the quantity $p_c(c', \varepsilon)$ under different priors. The goal here is to show the relation between the SED and the limit. Third, we reverse-engineer the theoretical results to find, if possible, data distributions satisfying $SED(p(\boldsymbol{h}|c), p(\boldsymbol{h}|c')) > SED(p(\boldsymbol{x}|c), p(\boldsymbol{x}|c'))$. There is no dataset associated with this experiment, rather we learn the parameters of the graphical model of Figure 1 (left) together with $\varepsilon, k, \sigma_\varepsilon$ ($k$ is treated as a continuous value during the optimization) in order to minimize the following objective: $\mathcal{L}_{SED} - \lambda \mathcal{L}_{CCNS}$, where $\lambda$ is a hyper-parameter. $\mathcal{L}_{SED}$ sums $SED(p(\boldsymbol{x}|c), p(\boldsymbol{x}|c')) - SED(p(\boldsymbol{h}|c), p(\boldsymbol{h}|c'))$ for all pairs of distinct classes $c, c'$, whereas $\mathcal{L}_{CCNS}$ computes the lower bound for the inter-class similarity, thus acting as a regularizer that avoids the trivial solution $p(\boldsymbol{x}|c) = p(\boldsymbol{x}|c')$ for class pairs $c, c'$. The set of configurations for this experiment is reported in Table 4, and we use Adam [35] as optimizer. In the last qualitative experiment, we compute the true CCNS and its approximations for the specific example of Figure 1 (right), showing that the quantities obtained in Proposition 3.3 and Theorem 3.5 are good representatives of the CCNS for nearest neighbor graphs. Please refer to Appendix A.5 for an additional study of this kind. [2]

## 5 Empirical Results

We now present our empirical results to support the theory. The interested reader can find statistics on the chosen hyper-parameters and additional ablation studies in Appendix A.6.

**Quantitative Results** Table 1 details the empirical comparison between the structure-agnostic MLP and the structure-aware baselines sDGN, GIN, and GCN. We observe that in terms of Micro F1 score, which is the metric all models are optimized against, there is *no statistically significant improvement* in performance (unpaired two samples t-test with p-value 0.05) for the different DGNs when using an artificial $k$-NN graph. When looking at accuracy one can observe a similar trend, but the results fluctuate more because accuracy is not the metric to be optimized. These results further confirm that using $k$-NN graphs does not bring a substantial contribution to the generalization performances. On the contrary, it would seem the $k$-NN graph can even be considered harmful for the performances of GIN and GCN (e.g., DryBean, Musk, eGrid, Waveform). We hypothesize that the information flow

---

[2]Code: `https://github.com/nec-research/class_distributions_graphs_tabular_data`

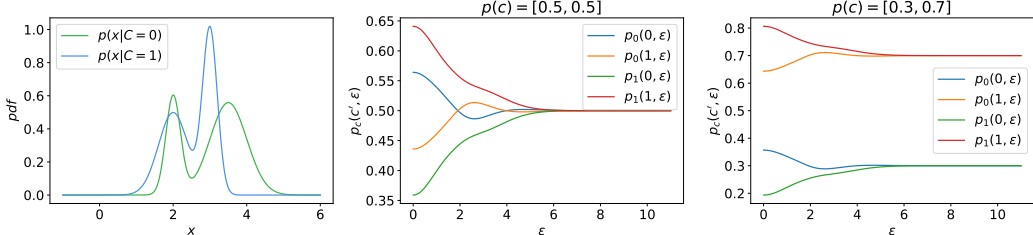

Figure 2: We show how $p_c(c', \varepsilon)$ varies for different values of $\varepsilon$ and prior distributions assuming the class-conditional data distributions on the left.

of message passing does not act anymore as a regularizer (we refer the reader to the discussion in Section 1) as it might happen the scarce supervision scenario [23], rather it becomes harmful noise for the learning process. This is possibly the reason why sDGN, which is also the simplest of the DGNs considered, tends to overfit less such noise and always performs on par with the MLP.

Therefore, one of the main takeaways of the quantitative analysis is that the $k$-NN graph might generally not be a good artificial structure for addressing the classification of tabular data using graph representation learning methods. Different artificial structures, possibly inspired by our theoretical results, should be proposed to create graphs that help DGNs to better generalize in this context. We note that this research direction is broader than tabular classification, but tabular data offers a great starting point since samples are typically assumed *i.i.d.* in ML and are thus easier to manipulate. Also, in Appendix A.4, we discuss the impact of a different $\Omega$ (the cosine similarity) in the construction of the $k$-NN graphs.

These results also substantiate our conjecture that, under assumptions $1, 3$, the $k$-NN graph *never* induces a better class separability in the embedding space: since $p(\boldsymbol{h}|c)$ is a mixture of distributions with the same mean but higher variance than $p(\boldsymbol{x}|c)$ (proof of Proposition 3.8) , in a low-dimensional space intuition suggests that the distributions of different classes will always overlap more in the embedding space than in the input space.

**Qualitative Results** We present the main qualitative results in Figures 2, 3 and 4. First, as we can see from Figure 2, when $\varepsilon$ is very small the probability that a neighbor is of the same class is high for both classes (but not the same), since there is less overlap between samples of the green ($C = 0$) and blue ($C = 1$) distributions. As we increase the side length of the hypercube (or equivalently, we consider more and more nearest neighbors), the curves converge to the priors that we set for this example. This result is in agreement with the second limit of Proposition 3.7.

As regards the first limit of Proposition 3.7, in Figure 3 we observe that the behavior of $\lim_{\varepsilon \to 0} p_c(c', \varepsilon)$ is non-trivial when the two curves still overlap. In particular, the greater the overlap the closer $p_c(c, \varepsilon)$ is to $p(c)$, which makes sense since it becomes harder to distinguish samples of different classes. On the other hand, the behavior becomes trivial when the curves are distant enough from each other: there, in the limit, every node will only have neighbors of the same class.

In the reverse-engineering experiment of Figure 4 (left), for each possible configuration of the models tried in Table 4, we have computed the $\mathcal{L}_{SED}$ for varying values of $k \in [1, 500]$. We have then normalized all values for readability, by dividing each value of $SED(p(\boldsymbol{h}|c), p(\boldsymbol{h}|c'))$ by $SED(p(\boldsymbol{x}|c), p(\boldsymbol{x}|c'))$, as the latter is independent of $k$. This particular figure depicts the curves for the binary classification cases, but the conclusions do not change for multi-class classification, for instance with 5 classes. We find that the normalized value of $SED(p(\boldsymbol{h}|c), p(\boldsymbol{h}|c'))$ on the $y$-axis is always upper-bounded by 1, meaning that $SED(p(\boldsymbol{h}|c), p(\boldsymbol{h}|c')) < SED(p(\boldsymbol{x}|c), p(\boldsymbol{x}|c'))$ for all the configurations tried, even in higher dimensions. This result further indicates that it could be unlikely that a $k$-NN graph induces better class separability in the embedding space of a DGN. We ask machine learning practitioners to take these insights into consideration when building artificial nearest neighbor graphs.

Lastly, we go back to the data distribution of Figure 1 (right) and show how to approximate the true CCNS under the $k$-NN graph with our theoretical framework. We first generate 10000 data points and set *w.l.o.g.* $k = 5$ to connect them. We compute the lower bound of the CCNS in the first

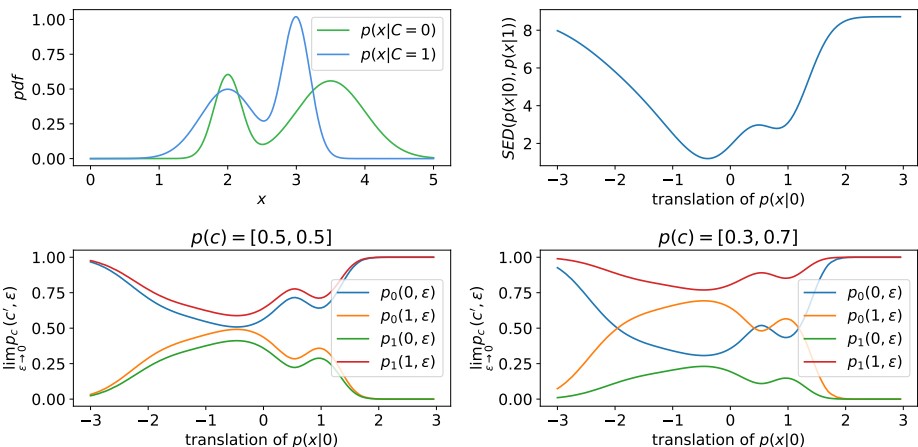

Figure 3: We show how $p_c(c', \varepsilon)$ varies for different translations of the conditional data distributions $p(x|0)$ and different priors as $\varepsilon$ approaches 0.

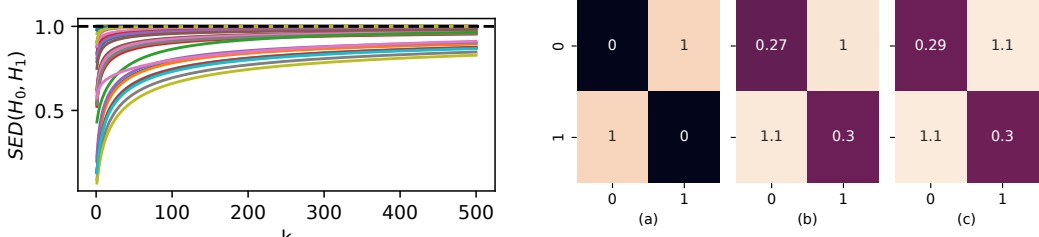

Figure 4: *Left:* Each curve specifies the variation of $SED(p(\boldsymbol{h}|c), p(\boldsymbol{h}|c'))$ against different values of $k$ for different hyperparameters' configurations that have been trained to minimize the learning objective. The curves are normalized w.r.t their corresponding $SED(p(\boldsymbol{x}|c), p(\boldsymbol{x}|c'))$. *Right:* The CCNS lower bound (a), its Monte Carlo approximation for $\varepsilon \approx 0.1$, obtained without creating any graph (b), and the true CCNS (c) for the example of Figure 1 and a 5-NN graph. Heatmaps (a) and (b) are computed using the theory of Section 3.1. Please refer to the text for more details.

heatmap (a), a Monte Carlo (MC) approximation of Equation 1 (1000 samples per class pair) in the heatmap (b), and the true CCNS in the heatmap (c). In this case the lower bound has computed a good approximation of the inter-class CCNS, and the MC estimate is very close to the true CCNS. In Appendix A.6, we also show that the MC approximation is better for higher values of $k$. It is important to recall that both heatmaps (a) and (b) are computed using the theoretical results of Section 3.1 and an $\varepsilon$ close to the empirical one, hence it seems that the theory is in accord with the evidence.

## 6 Conclusions

We introduced a new theoretical tool to understand how much nearest neighbor graphs can help in the classification of tabular data. Our theory and empirical evidence suggest that some attribute-based graph construction mechanisms, namely the $k$-NN algorithm, are not promising strategies for obtaining better DGNs' generalization performances. This is a particularly troubling result since researchers often used nearest neighbor graphs when the graph structure was unavailable. We recommend using great care in future empirical evaluations of such techniques by always comparing against structure-agnostic baselines to ascertain real improvements from fictitious ones. Moreover, we provided a theoretically principled way to model the CCNS for nearest neighbor graphs, showing its approximation in a practical example. We hope that the results of this work will foster better strategies for building artificial graphs or as a guideline for new structure learning methods.

## Acknowledgments and Disclosure of Funding

The author would like to acknowledge, in no particular order, Sascha Saralajew, Julia Gastinger, Ammar Shaker, Filippo Grazioli, Luca Oneto, Sandra Tartarelli, and Timo Sztyler for the stimulating discussions and extremely helpful feedback. A warm thank you also goes to the reviewers for striving to further improve this work and make it ready for publication.

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

# A Appendix

## A.1 Dataset Statistics

This section shows the main statistics of the datasets used in this work.

Table 2: Datasets' statistics.

|  | # samples | # attributes | # classes |
|---|---|---|---|
| Abalone | 4177 | 8 | 3 |
| Adult | 48842 | 14 | 2 |
| DryBean | 13611 | 17 | 7 |
| eGrid | 10000 | 14 | 2 |
| Isolet | 7797 | 617 | 26 |
| Musk | 6598 | 168 | 2 |
| Occupancy | 20560 | 7 | 2 |
| Waveform | 5000 | 40 | 3 |
| Cora | 2708 | 1433 | 7 |
| Citeseer | 3327 | 3703 | 6 |
| Pubmed | 19717 | 500 | 3 |

## A.2 Hyper-Parameters Tried

The table below specifies the configurations tried by the model selection procedure for all the baselines considered in this work. We apply a patience-based early stopping technique on the validation Micro F1-score as the only regularization technique [55]. The free parameters' budget of all three baselines is the same. We used Adam to train all models.

Table 3: Hyper-parameters tried during model selection for all the benchmarks of the paper. Note that in GCN the use of the normalized Laplacian implies a mean aggregation.

|  | # hidden units | # layers | learning rate | $k$ | epochs | patience | aggregation |
|---|---|---|---|---|---|---|---|
| MLP | 32/64/128 | 1/2/3 | 1e-2/1e-3 | - | 5000 | 1000 | - |
| sDGN | 32/64/128 | 1/2/3 | 1e-2/1e-3 | 1/3/5/10 | 5000 | 1000 | mean |
| GIN | 32/64/128 | 1/2/3 | 1e-2/1e-3 | 1/3/5/10 | 5000 | 1000 | sum/mean |
| GCN | 32/64/128 | 1/2/3 | 1e-2/1e-3 | 1/3/5/10 | 5000 | 1000 | mean |

The following table, instead, reports the configurations tried during the qualitative experiments. We still apply a patience-based early stopping technique on $\mathcal{L}_{SED}$.

Table 4: Hyper-parameters tried for the qualitative experiments.

| $D$ | $|\mathcal{M}|$ | $|\mathcal{C}|$ | $\lambda$ | learning rate | epochs | patience |
|---|---|---|---|---|---|---|
| 1, 2, 5, 10, 50 | 2, 5, 10 | 2, 5 | 0.0, 1.0, 5.0, 10.0 | 1e-2 | 10000 | 3000 |

### A.3 Proofs

**Proposition 3.3.** Under assumptions 1,2, $M_{\boldsymbol{x}}(c, \varepsilon)$ has the following analytical form

$$p(c) \sum_{m=1}^{|\mathcal{M}|} p(m|c) \prod_{f=1}^{D} \left( F_{Z_{cmf}} \left( x_f + \frac{\varepsilon}{2} \right) - F_{Z_{cmf}} \left( x_f - \frac{\varepsilon}{2} \right) \right), \tag{11}$$

where $Z_{cmf}$ is a *r.v.* with Gaussian distribution $\mathcal{N}(\cdot; \mu_{cmf}, \sigma_{cmf}^2)$.

*Proof.* Thanks to the conditional independence assumption, we can compute the integral of a product as a product of integrals over the independent dimensions. Defining $a_f = x_f - \frac{\varepsilon}{2}$ and $b_f = x_f + \frac{\varepsilon}{2}$, we have

$$\mathbb{E}_{\boldsymbol{w} \in H_\varepsilon(\boldsymbol{x})}[p(C = c|\boldsymbol{w})] = p(c) \int_{\boldsymbol{w} \in H_\varepsilon(\boldsymbol{x})} p(\boldsymbol{w}|c)d\boldsymbol{w} \tag{12}$$

$$= p(c) \int_{\boldsymbol{w} \in H_\varepsilon(\boldsymbol{x})} \sum_{m=1}^{|\mathcal{M}|} p(\boldsymbol{w}|m, c)p(m|c)d\boldsymbol{w} \tag{13}$$

$$= p(c) \sum_{m=1}^{|\mathcal{M}|} p(m|c) \int_{\boldsymbol{w} \in H_\varepsilon(\boldsymbol{x})} p(\boldsymbol{w}|m, c)d\boldsymbol{w} \tag{14}$$

$$= p(c) \sum_{m=1}^{|\mathcal{M}|} p(m|c) \int_{a_1}^{b_1} \cdot \cdot \int_{a_D}^{b_D} \prod_{f=1}^{D} p(w_f|m, c)dw_D \dots dw_1 \tag{15}$$

$$= p(c) \sum_{m=1}^{|\mathcal{M}|} p(m|c) \prod_{f=1}^{D} \int_{a_f}^{b_f} p(w_f|m, c)dw_f \tag{16}$$

$$= p(c) \sum_{m=1}^{|\mathcal{M}|} p(m|c) \prod_{f=1}^{D} \left( F_{Z_{cmf}} \left( x_f + \frac{\varepsilon}{2} \right) - F_{Z_{cmf}} \left( x_f - \frac{\varepsilon}{2} \right) \right), \tag{17}$$

where $Z_{cmf} \sim \mathcal{N}(\cdot; \mu_{cmf}, \sigma_{cmf}^2)$

and the last equality follows from the known fact $p(a \leq X \leq b) = F(b) - F(a)$. $\qquad \square$

We introduce the following lemmas that will be useful in Theorem 3.5 and Theorem 3.8, which are about addition, linear transformation, and marginalization involving mixtures of distributions.

**Lemma A.1.** *Let $X, Y$ be two independent random variables with corresponding mixture distributions $\phi_X(\boldsymbol{w}) = \sum_{i=1}^{I} \alpha_i f_i(\boldsymbol{w})$ and $\phi_Y(\boldsymbol{w}) = \sum_{j=1}^{J} \beta_j d_j(\boldsymbol{w})$. Then $Z = X + Y$ still follows a mixture distribution.*

*Proof.* By linearity of expectation the moment generating function of $X$ (and analogously $Y$) has the following form

$$M_X(\boldsymbol{t}) = \mathbb{E}[e^{\boldsymbol{t}^T \boldsymbol{w}}] = \int e^{\boldsymbol{t}^T \boldsymbol{w}} \phi_X(\boldsymbol{w})d\boldsymbol{w} \tag{18}$$

$$= \int e^{\boldsymbol{t}^T \boldsymbol{w}} \sum_{i=1}^{I} \alpha_i f_i(\boldsymbol{w})d\boldsymbol{w} \tag{19}$$

$$= \sum_{i=1}^{I} \alpha_i M_{X_i}(\boldsymbol{t}) \tag{20}$$

where $X_i$ is the *r.v.* corresponding to a component of the distribution. Using the fact that the moment generating function of $Z = X + Y$ is given by $M_Z(\boldsymbol{t}) = M_X(\boldsymbol{t})M_Y(\boldsymbol{t})$, we have that

$$M_Z(\boldsymbol{t}) = \left( \sum_{i=1}^{I} \alpha_i M_{X_i}(\boldsymbol{t}) \right) \left( \sum_{j=1}^{J} \beta_j M_{Y_j}(\boldsymbol{t}) \right) = \sum_{i=1}^{I} \sum_{j=1}^{J} \alpha_i \beta_j \underbrace{M_{X_i}(\boldsymbol{t})M_{Y_j}(\boldsymbol{t})}_{Z_{ij} = X_i + Y_j}. \tag{21}$$

Therefore, $Z$ follows a mixture model with $IJ$ components where each component follows the distribution associated with the random variable $Z_{ij} = X_i + Y_j$. $\square$

**Lemma A.2.** *Let $X$ be a r.v. with multivariate Gaussian mixture distribution $\phi_X(\boldsymbol{w}) = \sum_{i=1}^{I} \alpha_i \mathcal{N}(\boldsymbol{w}; \boldsymbol{\mu}_i, \boldsymbol{\Sigma}_i), \boldsymbol{w} \in \mathbb{R}^D$. Then $Z = \boldsymbol{\Lambda}X, \boldsymbol{\Lambda} \in \mathbb{R}^{D \times D}$ still follows a mixture distribution.*

*Proof.* Using the change of variable $\boldsymbol{z} = \boldsymbol{\Lambda}\boldsymbol{w}$ we have

$$\phi_Z(\boldsymbol{w}) = \phi_X(\boldsymbol{\Lambda}^{-1}\boldsymbol{w})|det(\boldsymbol{\Lambda}^{-1})| = \sum_{i=1}^{I} \alpha_i \mathcal{N}(\boldsymbol{\Lambda}^{-1}\boldsymbol{w}; \boldsymbol{\mu}_i, \boldsymbol{\Sigma}_i)|det(\boldsymbol{\Lambda}^{-1})|. \tag{22}$$

By expanding the terms, we see that the distribution of $Z$ is still a mixture of distributions of the following form

$$\phi_Z(\boldsymbol{w}) = \sum_{i=1}^{I} \alpha_i \mathcal{N}(\boldsymbol{w}; \boldsymbol{\Lambda}\boldsymbol{\mu}_i, \boldsymbol{\Lambda}\boldsymbol{\Sigma}_i\boldsymbol{\Lambda}^T). \tag{23}$$

$\square$

**Lemma A.3.** *Let $X, Y$ be two independent random variables with corresponding Gaussian mixtures of distributions $\phi_X(w) = \sum_{i=1}^{I} \alpha_i \mathcal{N}(w; \mu_{X_i}, \sigma_{X_i}^2)$ and $\phi_Y(w) = \sum_{j=1}^{J} \beta_j \mathcal{N}(w; \mu_{Y_j}, \sigma_{Y_j}^2)$. Then*

$$\mathbb{E}_X[F_Y(w)] = \int F_Y(w)\phi_X(w)dw = \sum_{i=1}^{I}\sum_{j=1}^{J} \alpha_i\beta_j \boldsymbol{\Phi}\left(\frac{\mu_{X_i} - \mu_{Y_j}}{\sqrt{\sigma_{Y_j}^2 + \sigma_{X_i}^2}}\right) \tag{24}$$

*Proof.* It is useful to look at this integral from a probabilistic point of view. In particular, we know that

$$p(Y \leq X | X = w) = p(Y \leq w) = F_Y(w) \tag{25}$$

and that, by marginalizing over all possible values of X,

$$p(Y \leq X) = \int p(Y \leq X | X = w)p(X = w)dw = \underbrace{\int F_Y(w)\phi_X(w)dw}_{\mathbb{E}_X[F_Y(w)]}. \tag{26}$$

Therefore, finding the solution corresponds to computing $p(Y - X \leq 0)$. Because $X, Y$ are independent, the resulting variable $Z = Y - X$ is distributed as (using Lemma A.1 and Lemma A.2)

$$p_Z(w) = \sum_{i=1}^{I}\sum_{j=1}^{J} \alpha_i\beta_j \mathcal{N}(w; \mu_{Y_j} - \mu_{X_i}; \sigma_{Y_j}^2 + \sigma_{X_i}^2) \tag{27}$$

and hence, using the fact that the *c.d.f.* of a mixture of Gaussians is the weighted sum of the individual components' *c.d.f.s*:

$$p(Z \leq 0) = \sum_{i=1}^{I}\sum_{j=1}^{J} \alpha_i\beta_j \boldsymbol{\Phi}\left(\frac{\mu_{X_i} - \mu_{Y_j}}{\sqrt{\sigma_{Y_j}^2 + \sigma_{X_i}^2}}\right). \tag{28}$$

$\square$

**Theorem 3.5.** Under assumptions 1,2, $M_{c'}(c, \varepsilon)$ has the following analytical form

$$p(c) \sum_{\substack{m=1 \\ m'=1}}^{|\mathcal{M}|} p(m'|c')p(m|c) \prod_{f=1}^{D} \left( \Phi \left( \frac{\mu_{c'm'f} + \frac{\varepsilon}{2} - \mu_{cmf}}{\sqrt{\sigma_{cmf}^2 + \sigma_{c'm'f}^2}} \right) - \Phi \left( \frac{\mu_{c'm'f} - \frac{\varepsilon}{2} - \mu_{cmf}}{\sqrt{\sigma_{cmf}^2 + \sigma_{c'm'f}^2}} \right) \right).$$

(29)

*Proof.* The proof relies on the previous Lemma A.3. We begin by expanding the formula and using the result of Proposition 3.3. We define the *r.v.* $Z_{imf} \sim \mathcal{N}(w; \mu_{imf}, \sigma_{imf}^2)$ to write

$$\mathbb{E}_{\boldsymbol{x} \sim p(\boldsymbol{x}|c')}[M_{\boldsymbol{x}}(c)] \tag{30}$$

$$= \int p(c) \sum_{m=1}^{|\mathcal{M}|} p(m|c) \prod_{f=1}^{D} \left( F_{Z_{cmf}} \left( x_f + \frac{\varepsilon}{2} \right) - F_{Z_{cmf}} \left( x_f - \frac{\varepsilon}{2} \right) \right) p(\boldsymbol{x}|c') d\boldsymbol{x} \tag{31}$$

$$= p(c) \int \left( \sum_{m=1}^{|\mathcal{M}|} p(m|c) \prod_{f=1}^{D} \left( F_{Z_{cmf}} \left( x_f + \frac{\varepsilon}{2} \right) - F_{Z_{cmf}} \left( x_f - \frac{\varepsilon}{2} \right) \right) \right) \left( \sum_{m'=1}^{|\mathcal{M}|} p(\boldsymbol{x}|m', c')p(m'|c') \right) d\boldsymbol{x} \tag{32}$$

$$p(c) \int \sum_{\substack{m=1 \\ m'=1}}^{|\mathcal{M}|} p(m'|c')p(m|c) \prod_{f=1}^{D} \left( F_{Z_{cmf}} \left( x_f + \frac{\varepsilon}{2} \right) - F_{Z_{cmf}} \left( x_f - \frac{\varepsilon}{2} \right) \right) p(\boldsymbol{x}|m', c') d\boldsymbol{x} \tag{33}$$

$$= p(c) \sum_{\substack{m=1 \\ m'=1}}^{|\mathcal{M}|} p(m'|c')p(m|c) \int \prod_{f=1}^{D} \left( F_{Z_{cmf}} \left( x_f + \frac{\varepsilon}{2} \right) - F_{Z_{cmf}} \left( x_f - \frac{\varepsilon}{2} \right) \right) \prod_{f=1}^{D} p(x_f|m', c') d\boldsymbol{x} \tag{34}$$

$$= p(c) \sum_{\substack{m=1 \\ m'=1}}^{|\mathcal{M}|} p(m'|c')p(m|c) \int \cdot \cdot \int \prod_{f=1}^{D} \left( F_{Z_{cmf}} \left( x_f + \frac{\varepsilon}{2} \right) - F_{Z_{cmf}} \left( x_f - \frac{\varepsilon}{2} \right) \right) p(x_f|m', c') d\boldsymbol{x} \tag{35}$$

$$= p(c) \sum_{\substack{m=1 \\ m'=1}}^{|\mathcal{M}|} p(m'|c')p(m|c) \prod_{f=1}^{D} \int \left( F_{Z_{cmf}} \left( x_f + \frac{\varepsilon}{2} \right) - F_{Z_{cmf}} \left( x_f - \frac{\varepsilon}{2} \right) \right) p(x_f|m', c') dx_f \tag{36}$$

$$= p(c) \sum_{\substack{m=1 \\ m'=1}}^{|\mathcal{M}|} p(m'|c')p(m|c) \times$$

$$\times \prod_{f=1}^{D} \left( \underbrace{\int F_{Z_{cmf}} \left( x_f + \frac{\varepsilon}{2} \right) p(x_f|m', c') dx_f}_{\mathbb{E}_{Z_{c'f}}[F_{Z_{cmf}}(x_f + \frac{\varepsilon}{2})]} - \underbrace{\int F_{Z_{cmf}} \left( x_f - \frac{\varepsilon}{2} \right) p(x_f|m', c') dx_f}_{\mathbb{E}_{Z_{c'f}}[F_{Z_{cmf}}(x_f - \frac{\varepsilon}{2})]} \right). \tag{37}$$

We first note that $F_{Z_{cmf}} \left( x_f + \frac{\varepsilon}{2} \right) = F_Y(x_f)$ where $Y$ follows distribution $\mathcal{N}(w; \mu_{cmf} - \frac{\varepsilon}{2}, \sigma_{cmf}^2)$ (and symmetrically for $F_{Z_{cmf}} \left( x_f - \frac{\varepsilon}{2} \right)$), so we can apply Lemma A.3 and obtain

$$p(c) \sum_{\substack{m=1 \\ m'=1}}^{|\mathcal{M}|} p(m'|c')p(m|c) \prod_{f=1}^{D} \left( \Phi \left( \frac{\mu_{c'm'f} + \frac{\varepsilon}{2} - \mu_{cmf}}{\sqrt{\sigma_{cmf}^2 + \sigma_{c'm'f}^2}} \right) - \Phi \left( \frac{\mu_{c'm'f} - \frac{\varepsilon}{2} - \mu_{cmf}}{\sqrt{\sigma_{cmf}^2 + \sigma_{c'm'f}^2}} \right) \right).$$

(38)

$\square$

**Proposition 3.7.** *Let the first $D$ derivatives of $\sum_{i=1}^{|\mathcal{C}|} M_{c'}(i)$ be different from 0 in an open interval $I$ around $\varepsilon = 0$ and not tending to infinity for $\varepsilon \to 0$. Then, under assumptions 1,2, Equation 7 has the following limits (the first of which requires the assumptions)*

$$\lim_{\varepsilon \to 0} p_{c'}(c, \varepsilon) = \frac{p(c) \sum_{\substack{m=1 \\ m'=1}}^{|\mathcal{M}|} p(m|c)p(m'|c') \prod_{f=1}^{D} \phi_{Z_{cmm'f}}(0)}{\sum_i^{|\mathcal{C}|} p(i) \sum_{\substack{m=1 \\ m'=1}}^{|\mathcal{M}|} p(m|c)p(m'|c') \prod_{f=1}^{D} \phi_{Z_{imm'f}}(0)} \qquad \lim_{\varepsilon \to \infty} p_{c'}(c, \varepsilon) = p(c)$$

$$(39)$$

where $Z_{imm'f}$ has distribution $\mathcal{N}(\cdot; -a_{imm'f}, b_{imm'f}^2)$,

$a_{imm'f} = 2(\mu_{c'm'f} - \mu_{imf})$ and $b_{imm'f} = 2\sqrt{\sigma_{imf}^2 + \sigma_{c'm'f}^2}$.

*Proof.* The second limit follows from $\lim_{x \to +\infty} \mathbf{\Phi}(x) = 1$ and $\lim_{x \to -\infty} \mathbf{\Phi}(x) = 0$. As regards the first limit, we first expand the terms

$$\lim_{\varepsilon \to 0} p_{c'}(c, \varepsilon) = \lim_{\varepsilon \to 0} \frac{p(c) \sum_{\substack{m=1 \\ m'=1}}^{|\mathcal{M}|} p(m|c)p(m'|c') \prod_{f=1}^{D} \left( \mathbf{\Phi}(\frac{\varepsilon + a_{cmm'f}}{b_{cmm'f}}) - \mathbf{\Phi}(\frac{-\varepsilon + a_{cmm'f}}{b_{cmm'f}}) \right)}{\sum_i^{|\mathcal{C}|} p(i) \sum_{\substack{m=1 \\ m'=1}}^{|\mathcal{M}|} p(m|i)p(m'|c') \prod_{f=1}^{D} \left( \mathbf{\Phi}(\frac{\varepsilon + a_{imm'f}}{b_{imm'f}}) - \mathbf{\Phi}(\frac{-\varepsilon + a_{imm'f}}{b_{imm'f}}) \right)},$$

$$(40)$$

where $a_{imm'f} = 2(\mu_{c'm'f} - \mu_{imf})$ and $b_{imm'f} = 2\sqrt{\sigma_{imf}^2 + \sigma_{c'm'f}^2} \quad \forall i \in \mathcal{C}$ to simplify the notation. By defining $Z_{imm'f} \sim \mathcal{N}(\cdot; -a_{imm'f}, b_{imm'f}^2)$, we can rewrite the limit as

$$\lim_{\varepsilon \to 0} \frac{p(c) \sum_{\substack{m=1 \\ m'=1}}^{|\mathcal{M}|} p(m|c)p(m'|c') \prod_{f=1}^{D} F_{Z_{cmm'f}}(\varepsilon) - F_{Z_{cmm'f}}(-\varepsilon)}{\sum_i^{|\mathcal{C}|} p(i) \sum_{\substack{m=1 \\ m'=1}}^{|\mathcal{M}|} p(m|i)p(m'|c') \prod_{f=1}^{D} F_{Z_{imm'f}}(\varepsilon) - F_{Z_{imm'f}}(-\varepsilon)} \qquad (41)$$

$$= \lim_{\varepsilon \to 0} \frac{p(c) \sum_{\substack{m=1 \\ m'=1}}^{|\mathcal{M}|} p(m|c)p(m'|c') \prod_{f=1}^{D} d_{cmm'f}(\varepsilon)}{\sum_i^{|\mathcal{C}|} p(i) \sum_{\substack{m=1 \\ m'=1}}^{|\mathcal{M}|} p(m|i)p(m'|c') \prod_{f=1}^{D} d_{imm'f}(\varepsilon)}, \qquad (42)$$

Both the numerator and denominator tend to zero in the limit, so we try to apply L'Hôpital's rule, potentially multiple times.

For reasons that will become clear soon, we show that the limit of the first derivative of each term in the product is not zero nor infinity:

$$\lim_{\varepsilon \to 0} d_{imm'f}^{(1)}(\varepsilon) = \lim_{\varepsilon \to 0} \left( \phi_{Z_{imm'f}}(\varepsilon) + \phi_{Z_{imm'f}}(-\varepsilon) \right) = 2\phi_{Z_{imm'f}}(0), \qquad (43)$$

In addition, let us consider the $n$-th derivative of the product of functions (by applying the generalized product rule):

$$\left( \prod_{f=1}^{D} d_{imm'f}(\varepsilon) \right)^{(n)} = \sum_{j_1 + \cdots + j_D = n} \binom{n}{j_1, \ldots, j_D} \prod_{f=1}^{D} d_{imm'f}^{(j_f)}(\varepsilon) \qquad (44)$$

and we note that, as long as $n < D$ there will exist an assignment to $j_f = 0 \; \forall f$ in the summation, and thus the limit of each product will always tend to 0 when $\varepsilon$ goes to 0 since the individual terms cannot tend to infinity for our assumptions. However, when we take the $D$-th derivative, there exists one term in the summation that does not go to 0 in the limit, which is

$$\binom{D}{1, \ldots, 1} \prod_{f=1}^{D} d_{imm'f}^{(1)}(\varepsilon) = D! \prod_{f=1}^{D} d_{imm'f}^{(1)}(\varepsilon) \qquad (45)$$

$$\lim_{\varepsilon \to 0} \binom{D}{1, \ldots, 1} \prod_{f=1}^{D} d_{imm'f}^{(1)}(\varepsilon) = 2^D D! \prod_{f=1}^{D} \phi_{Z_{imm'f}}(0)(\varepsilon) \qquad (46)$$

Therefore, by applying the L'Hôpital's rule $D$ times, we obtain

$$\lim_{\varepsilon \to 0} \frac{p(c) \sum_{\substack{m=1 \\ m'=1}}^{|\mathcal{M}|} p(m|c)p(m'|c') \prod_{f=1}^{D} d_{cmm'f}(\varepsilon)}{\sum_{i}^{|\mathcal{C}|} p(i) \sum_{\substack{m=1 \\ m'=1}}^{|\mathcal{M}|} p(m|i)p(m'|c') \prod_{f=1}^{D} d_{imm'f}(\varepsilon)} \tag{47}$$

$$= \frac{p(c) \sum_{\substack{m=1 \\ m'=1}}^{|\mathcal{M}|} p(m|c)p(m'|c') \prod_{f=1}^{D} \phi_{Z_{cmm'f}}(0)}{\sum_{i}^{|\mathcal{C}|} p(i) \sum_{\substack{m=1 \\ m'=1}}^{|\mathcal{M}|} p(m|i)p(m'|c') \prod_{f=1}^{D} \phi_{Z_{imm'f}}(0)}. \tag{48}$$

However, to be valid, L'Hôpital's rule requires that the derivative of the denominator never goes to 0 for points different from $\varepsilon = 0$. This is part of our assumptions, hence we conclude the proof. $\square$

For $n = 1$ we can show that this last requirement holds as $d^{(1)}_{imm'f}(\varepsilon) > 0 \; \forall \varepsilon$, $d_{imm'f}(\varepsilon) \geq 0$, and $d_{imm'f}(\varepsilon) = 0$ if and only if $\varepsilon = 0$. In fact,

$$\Big( \prod_{f=1}^{D} d_{imm'f}(\varepsilon) \Big)^{(1)} = \sum_{j_1 + \cdots + j_D = 1} \prod_{f=1}^{D} d^{(j_f)}_{imm'f}(\varepsilon) \tag{49}$$

is a sum over terms that are all greater than 0 for $\varepsilon \neq 0$.

**Analytical form of the $n$-th derivative of $\sum_{i=1}^{|\mathcal{C}|} M_{c'}(i)$** To verify if the hypotheses of Proposition 3.7 is true given the parameters of all features' distributions, one could compute the $n$-th derivative of the denominator w.r.t. $\varepsilon$ and check that it is not zero around 0. We start again from

$$\Big( \prod_{f=1}^{D} d_{imm'f}(\varepsilon) \Big)^{(n)} = \sum_{j_1 + \cdots + j_D = n} \binom{n}{j_1, \ldots, j_D} \prod_{f=1}^{D} d^{(j_f)}_{imm'f}(\varepsilon) \tag{50}$$

and we proceed using the fact that the $n$-th derivative of the standard normal distribution $S \sim \mathcal{N}(0,1)$ has a well-known form in terms of the $n$-th (probabilist) Hermite polynomial $He_n(x)$. We note the following equivalence:

$$\phi_{Z_{imm'f}}(w) = \frac{1}{b_{imm'f}} \phi_S\Big( \frac{w + a_{imm'f}}{b_{imm'f}} \Big), \tag{51}$$

hence we can write

$$d^{(n)}_{imm'f}(\varepsilon) = (d^{(1)}_{imm'f}(\varepsilon))^{(n-1)} = (\phi_{Z_{imm'f}}(\varepsilon) + \phi_{Z_{imm'f}}(-\varepsilon))^{(n-1)} \tag{52}$$

$$= (\phi_{Z_{imm'f}}(\varepsilon))^{(n-1)} + (\phi_{Z_{imm'f}}(-\varepsilon))^{(n-1)} \tag{53}$$

$$= \Big( \frac{1}{b_{imm'f}} \phi_S\Big( \frac{\varepsilon + a_{imm'f}}{b_{imm'f}} \Big) \Big)^{(n-1)} \tag{54}$$

$$+ \Big( \frac{1}{b_{imm'f}} \phi_S\Big( \frac{-\varepsilon + a_{imm'f}}{b_{imm'f}} \Big) \Big)^{(n-1)} \tag{55}$$

$$= \frac{1}{b_{imm'f}} \Big( (\phi_S\Big( \frac{\varepsilon + a_{imm'f}}{b_{imm'f}} \Big))^{(n-1)} + (\phi_S\Big( \frac{-\varepsilon + a_{imm'f}}{b_{imm'f}} \Big))^{(n-1)} \Big) \tag{56}$$

$$= \frac{1}{b_{imm'f}} \Big( (\mathcal{N}\Big( \frac{\varepsilon + a_{imm'f}}{b_{imm'f}}; 0, 1 \Big))^{(n-1)} + (\mathcal{N}\Big( \frac{-\varepsilon + a_{imm'f}}{b_{imm'f}}; 0, 1 \Big))^{(n-1)} \Big) \tag{57}$$

$$= \frac{1}{b_{imm'f}} \Big( (-1)^{n-1} He_{n-1}\Big( \frac{\varepsilon + a_{imm'f}}{b_{imm'f}} \Big) \phi_S\Big( \frac{\varepsilon + a_{imm'f}}{b_{imm'f}} \Big) \Big) +$$

$$+ (-1)^{n-1} He_{n-1}\Big( \frac{-\varepsilon + a_{imm'f}}{b_{imm'f}} \Big) \phi_S\Big( \frac{-\varepsilon + a_{imm'f}}{b_{imm'f}} \Big) \Big). \tag{58}$$

We use this result in the expansion of Equation 50

$$\sum_{j_1+\cdots+j_D=n} \binom{n}{j_1,\ldots,j_D} \prod_{f=1}^{D} d_{imm'f}^{(j_f)}(\varepsilon) \tag{59}$$

$$= \sum_{j_1+\cdots+j_D=n} \binom{n}{j_1,\ldots,j_D} \prod_{\substack{j_f>0 \\ f=1\ldots D}} d_{imm'f}^{(j_f)}(\varepsilon) \prod_{\substack{j_f=0 \\ f=1\ldots D}} d_{imm'f}(\varepsilon). \tag{60}$$

However, we can readily see that computing the derivative for a single $\varepsilon$ has combinatorial complexity, which makes the application of the above formulas practical only for small values of $D$.

**Class Separability Analysis of Section 3.2**   We now present a result that will help us compute the SED divergence between two Gaussian mixtures of distributions.

**Lemma A.4.** *Let $X, Y$ be two independent random variables with corresponding Gaussian mixture distributions $\phi_X(\boldsymbol{w}) = \sum_{i=1}^{I} \alpha_i f_{X_i}(\boldsymbol{w}) = \sum_{i=1}^{I} \alpha_i \mathcal{N}(\boldsymbol{w}; \boldsymbol{\mu}_i^X, \boldsymbol{\Sigma}_i^X)$ and $\phi_Y(\boldsymbol{w}) = \sum_{j=1}^{J} \beta_j f_{Y_j}(\boldsymbol{w}) = \sum_{j=1}^{J} \beta_j \mathcal{N}(\boldsymbol{w}; \boldsymbol{\mu}_j^Y, \boldsymbol{\Sigma}_j^Y)$, $\boldsymbol{w} \in \mathbb{R}^D$. Then the SED divergence between $\phi_X(\boldsymbol{w})$ and $\phi_Y(\boldsymbol{w})$ can be computed as*

$$SED(\phi_X, \phi_Y) = \sum_{i=1}^{I} \sum_{j=1}^{I} \alpha_i \alpha_j A_{i,j} + \sum_{i=1}^{J} \sum_{j=1}^{J} \beta_i \beta_j B_{i,j} - 2 \sum_{i=1}^{I} \sum_{j=1}^{J} \alpha_i \beta_j C_{i,j}, \tag{61}$$

*where* \tag{62}

$$A_{i,j} = \mathcal{N}(\boldsymbol{\mu}_i^X; \boldsymbol{\mu}_j^X, (\boldsymbol{\Sigma}_i^X + \boldsymbol{\Sigma}_j^X)) \tag{63}$$

$$B_{i,j} = \mathcal{N}(\boldsymbol{\mu}_i^Y; \boldsymbol{\mu}_j^Y, (\boldsymbol{\Sigma}_i^Y + \boldsymbol{\Sigma}_j^Y)) \tag{64}$$

$$C_{i,j} = \mathcal{N}(\boldsymbol{\mu}_i^X; \boldsymbol{\mu}_j^Y, (\boldsymbol{\Sigma}_i^X + \boldsymbol{\Sigma}_j^Y)). \tag{65}$$

*Proof.*

$$SED(\phi_X, \phi_Y) = \int (\phi_X(\boldsymbol{w}) - \phi_Y(\boldsymbol{w}))^2 d\boldsymbol{w} \tag{66}$$

$$SED(\phi_X, \phi_Y) = \int \left( \sum_{i=1}^{I} \alpha_i f_{X_i}(\boldsymbol{w}) - \sum_{j=1}^{J} \beta_j f_{Y_j}(\boldsymbol{w}) \right)^2 d\boldsymbol{w} \tag{67}$$

$$= \sum_{i=1}^{I} \sum_{j=1}^{I} \alpha_i \alpha_j \int f_{X_i}(\boldsymbol{w}) f_{X_j}(\boldsymbol{w}) d\boldsymbol{w} + \sum_{i=1}^{J} \sum_{j=1}^{J} \beta_i \beta_j \int f_{Y_i}(\boldsymbol{w}) f_{Y_j}(\boldsymbol{w}) d\boldsymbol{w} \tag{68}$$

$$- 2 \sum_{i=1}^{I} \sum_{j=1}^{J} \alpha_i \beta_j \int f_{X_i}(\boldsymbol{w}) f_{Y_j}(\boldsymbol{w}) d\boldsymbol{w}. \tag{69}$$

Finally, we can compute the integral of the product of two Gaussians as follows (Section 8.1.8 of [53])

$$\int \mathcal{N}(\boldsymbol{w}; \boldsymbol{\mu_1}, \boldsymbol{\Sigma_1}) \mathcal{N}(\boldsymbol{w}; \boldsymbol{\mu_2}, \boldsymbol{\Sigma_2}) d\boldsymbol{w} = \mathcal{N}(\boldsymbol{\mu_1}; \boldsymbol{\mu_2}, (\boldsymbol{\Sigma_1} + \boldsymbol{\Sigma_2})) \tag{70}$$

to obtain the desired value.   $\square$

**Proposition 3.8.**   Under assumptions 1,3, let us assume that the entity embeddings of a 1-layer DGN applied on an artificially built $k$-NN graph follow the distribution of Equation 10. Then the Squared Error Distances $SED(p(\boldsymbol{h}|c), p(\boldsymbol{h}|c'))$ and $SED(p(\boldsymbol{x}|c), p(\boldsymbol{x}|c'))$ have analytical forms.

*Proof.* First, we have to show how we can compute the conditional probabilities $p(\boldsymbol{h}|c)$ and $p(\boldsymbol{h}|c')$, and then we derive the analytical computation of the $SED$ divergence to verify the inequality in a similar manner of Helén and Virtanen [33].

We first work out the explicit form of $p(\boldsymbol{h}|c)$, by marginalizing out $\boldsymbol{x}$:

$$p(\boldsymbol{h}|c) = \int \sum_{m}^{|\mathcal{M}|} p(\boldsymbol{h}|\boldsymbol{x})p(\boldsymbol{x}|m,c)p(m|c)d\boldsymbol{x} \tag{71}$$

$$= \sum_{m}^{|\mathcal{M}|} p(m|c) \int p(\boldsymbol{h}|\boldsymbol{x})p(\boldsymbol{x}|m,c)d\boldsymbol{x} \tag{72}$$

$$= \sum_{m}^{|\mathcal{M}|} p(m|c) \int \mathcal{N}(\boldsymbol{h};\boldsymbol{x},\boldsymbol{\Lambda}_{\varepsilon})\mathcal{N}(\boldsymbol{x};\boldsymbol{\mu}_{mc},\boldsymbol{\Sigma}_{mc})d\boldsymbol{x} \tag{73}$$

$$= \sum_{m}^{|\mathcal{M}|} p(m|c) \underbrace{\int \mathcal{N}(\boldsymbol{h}-\boldsymbol{x};\boldsymbol{0},\boldsymbol{\Lambda}_{\varepsilon})\mathcal{N}(\boldsymbol{x};\boldsymbol{\mu}_{mc},\boldsymbol{\Sigma}_{mc})d\boldsymbol{x}}_{\text{Gaussian convolution}} \tag{74}$$

$$= \sum_{m}^{|\mathcal{M}|} p(m|c)\mathcal{N}(\boldsymbol{h};\boldsymbol{\mu}_{mc},\boldsymbol{\Sigma}_{mc}+\boldsymbol{\Lambda}_{\varepsilon}). \tag{75}$$

Therefore, the distribution resulting from the $k$-NN neighborhood aggregation will change the input's variance in a way that is inversely proportional to the number of neighbors. In the limit of $k \to \infty$ and finite $\sigma_{\varepsilon}$, it follows that $p(\boldsymbol{h}|c) = p(\boldsymbol{x}|c)$. Since $p(\boldsymbol{h}|c)$ still follows a mixture of distributions of known form, we note that a repeated aggregation of the neighborhood aggregation mechanism (without any learned transformation) would only increase the values of the covariance matrix. In turn, this would make the distribution spread more and more, causing what is known in the literature as the *oversmoothing* effect (see Section 2).

Since both $X_c$ and $H_c$ follow a Gaussian mixture distribution, we apply Lemma A.4 to obtain a closed-form solution and be able to evaluate the inequality. In particular

$$SED(\phi_{X_c},\phi_{X_{c'}}) = \tag{76}$$

$$= \sum_{\substack{m=1 \\ m'=1}}^{|\mathcal{M}|} p(m|c)p(m'|c)\mathcal{N}(\boldsymbol{\mu}_{cm};\boldsymbol{\mu}_{cm'},\boldsymbol{\Sigma}_{cm}+\boldsymbol{\Sigma}_{cm'}) \tag{77}$$

$$+ \sum_{\substack{m=1 \\ m'=1}}^{|\mathcal{M}|} p(m|c')p(m'|c')\mathcal{N}(\boldsymbol{\mu}_{c'm};\boldsymbol{\mu}_{c'm'},\boldsymbol{\Sigma}_{c'm}+\boldsymbol{\Sigma}_{c'm'}) \tag{78}$$

$$- 2\sum_{\substack{m=1 \\ m'=1}}^{|\mathcal{M}|} p(m|c)p(m'|c')\mathcal{N}(\boldsymbol{\mu}_{cm};\boldsymbol{\mu}_{c'm'},\boldsymbol{\Sigma}_{cm}+\boldsymbol{\Sigma}_{c'm'}). \tag{79}$$

and

$$SED(\phi_{H_c},\phi_{H_{c'}}) = \tag{80}$$

$$= \sum_{\substack{m=1 \\ m'=1}}^{|\mathcal{M}|} p(m|c)p(m'|c)\mathcal{N}(\boldsymbol{\mu}_{cm};\boldsymbol{\mu}_{cm'},\boldsymbol{\Sigma}_{cm}+\boldsymbol{\Sigma}_{cm'}+2\boldsymbol{\Lambda}_{\varepsilon}) \tag{81}$$

$$+ \sum_{\substack{m=1 \\ m'=1}}^{|\mathcal{M}|} p(m|c')p(m'|c')\mathcal{N}(\boldsymbol{\mu}_{c'm};\boldsymbol{\mu}_{c'm'},\boldsymbol{\Sigma}_{c'm}+\boldsymbol{\Sigma}_{c'm'}+2\boldsymbol{\Lambda}_{\varepsilon}) \tag{82}$$

$$- 2\sum_{\substack{m=1 \\ m'=1}}^{|\mathcal{M}|} p(m|c)p(m'|c')\mathcal{N}(\boldsymbol{\mu}_{cm};\boldsymbol{\mu}_{c'm'},\boldsymbol{\Sigma}_{cm}+\boldsymbol{\Sigma}_{c'm'}+2\boldsymbol{\Lambda}_{\varepsilon}). \tag{83}$$

$\square$

As an additional remark, if $SED(\phi_{H_c}, \phi_{H_{c'}}) > SED(\phi_{X_c}, \phi_{X_{c'}})$, points in the DGN's latent space will be easier to classify than those in the input space because the former pair of distributions are more separable than the latter.

We also observe that $p(\boldsymbol{h}|c)$ is a mixture of Gaussians that differs from $p(\boldsymbol{x}|c)$ only in the higher variance of the individual Gaussian components. In other words, the distribution of $p(\boldsymbol{h}|c)$ looks "flatter" than $p(\boldsymbol{x}|c)$. At least in a one or two-dimensional space, flatter distributions $p(\boldsymbol{h}|c)$ and $p(\boldsymbol{h}|c')$ should therefore increase the overlap (and therefore decrease the SED) compared to the SED in the original space. This intuition, despite its simplicity, is unfortunately hard to prove for technical reasons, for instance the dependence of the distance value on the specific parameters $\mu$ and $\Sigma$.

### A.4 Impact of the Metric in the Construction of $k$-NN Graphs

To judge how much the results might change if we used a different metric, we test the cosine similarity as done in [56]. We note that the cosine distance is less expressive than the Euclidean distance as it does not consider the magnitude of values, and that results in overall worse performances. DGNs typically work in the Euclidean space when aggregating neighbors, hence the homophily assumption can be imposed - to a certain extent - only by computing nearest neighbors in terms of Euclidean distance. In addition, building $k$-NN graphs using the Euclidean distance criterion has an easy interpretation in terms of "how similar the attributes of adjacent nodes are". We observe that the results below still agree with our theoretical intuitions and empirical results.

Table 5: Node classification mean and standard deviation results for a structure-agnostic baseline (MLP) compared to graph machine learning methods applied to artificially built $k$-NN graphs. We performed model selection w.r.t. the F1 score, but we also include the accuracy for completeness. Here, $k$-NN graphs are built using the *cosine similarity* as metric.

| | MLP | | sDGN | | GIN | | GCN | |
|---|---|---|---|---|---|---|---|---|
| | F1 | ACC | F1 | ACC | F1 | ACC | F1 | ACC |
| Abalone | $0.55_{\pm 0.02}$ | $55.5_{\pm 1.5}$ | $0.55_{\pm 0.01}$ | $55.5_{\pm 0.9}$ | $0.55_{\pm 0.03}$ | $55.2_{\pm 1.8}$ | $0.55_{\pm 0.02}$ | 55.1 (1.9) |
| Adult | $0.70_{\pm 0.02}$ | $77.6_{\pm 1.7}$ | $0.64_{\pm 0.01}$ | $79.0_{\pm 0.6}$ | $0.69_{\pm 0.03}$ | $79.5_{\pm 0.6}$ | $0.53_{\pm 0.07}$ | 76.4 (0.6) |
| DryBean | $0.78_{\pm 0.04}$ | $77.4_{\pm 4.0}$ | $0.77_{\pm 0.06}$ | $77.0_{\pm 5.0}$ | $0.79_{\pm 0.02}$ | $78.2_{\pm 2.0}$ | $0.78_{\pm 0.06}$ | 78.2 (5.3) |
| eGrid | $0.95_{\pm 0.01}$ | $95.6_{\pm 0.6}$ | $0.84_{\pm 0.01}$ | $85.8_{\pm 0.9}$ | $0.93_{\pm 0.01}$ | $93.9_{\pm 0.8}$ | $0.81_{\pm 0.01}$ | 82.6 (1.4) |
| Isolet | $0.95_{\pm 0.01}$ | $95.4_{\pm 0.5}$ | $0.93_{\pm 0.01}$ | $92.7_{\pm 1.0}$ | $0.95_{\pm 0.01}$ | $94.8_{\pm 1.0}$ | $0.90_{\pm 0.05}$ | 90.6 (4.1) |
| Musk | $0.99_{\pm 0.01}$ | $99.3_{\pm 0.3}$ | $0.96_{\pm 0.01}$ | $98.0_{\pm 0.6}$ | $0.93_{\pm 0.04}$ | $96.2_{\pm 2.5}$ | $0.93_{\pm 0.04}$ | 97.1 (0.7) |
| Occupancy | $0.99_{\pm 0.01}$ | $98.9_{\pm 0.4}$ | $0.99_{\pm 0.01}$ | $98.6_{\pm 0.3}$ | $0.99_{\pm 0.01}$ | $98.9_{\pm 0.3}$ | $0.95_{\pm 0.1}$ | 97.6 (4.8) |
| Waveform | $0.85_{\pm 0.02}$ | $85.5_{\pm 1.7}$ | $0.84_{\pm 0.01}$ | $84.0_{\pm 1.4}$ | $0.84_{\pm 0.01}$ | $84.1_{\pm 1.6}$ | $0.82_{\pm 0.03}$ | 82.6 (2.3) |
| Cora | $0.71_{\pm 0.04}$ | $74.3_{\pm 2.4}$ | $0.71_{\pm 0.03}$ | $74.1_{\pm 2.7}$ | $0.72_{\pm 0.02}$ | $75.5_{\pm 1.8}$ | $0.72_{\pm 0.03}$ | 75.2 (2.3) |
| Citeseer | $0.69_{\pm 0.02}$ | $71.8_{\pm 1.9}$ | $0.69_{\pm 0.02}$ | $71.8_{\pm 1.8}$ | $0.70_{\pm 0.02}$ | $73.7_{\pm 1.3}$ | $0.69_{\pm 0.02}$ | 71.7 (2.4) |
| Pubmed | $0.87_{\pm 0.01}$ | $87.6_{\pm 0.7}$ | $0.84_{\pm 0.01}$ | $83.8_{\pm 0.7}$ | $0.86_{\pm 0.01}$ | $86.7_{\pm 0.8}$ | $0.84_{\pm 0.01}$ | 83.7 (0.6) |

## A.5 Analysis of CCNS approximation's quality

In the figure below, we visualize how the true CCNS gets closer to the MC approximation when the number of neighbors $k$ increases for a random data distribution with 5 classes. The choice of the $\varepsilon$ parameter for the MC estimation of the CCNS has a marginal impact here. This behavior is expected, because as $k$ increases we get a better approximation of the posterior class distribution in the hypercube around each point, which is the one we computed in Section 3.1. When $k$ is too small, the empirical (true) CCNS might be more unstable and dataset-dependent.

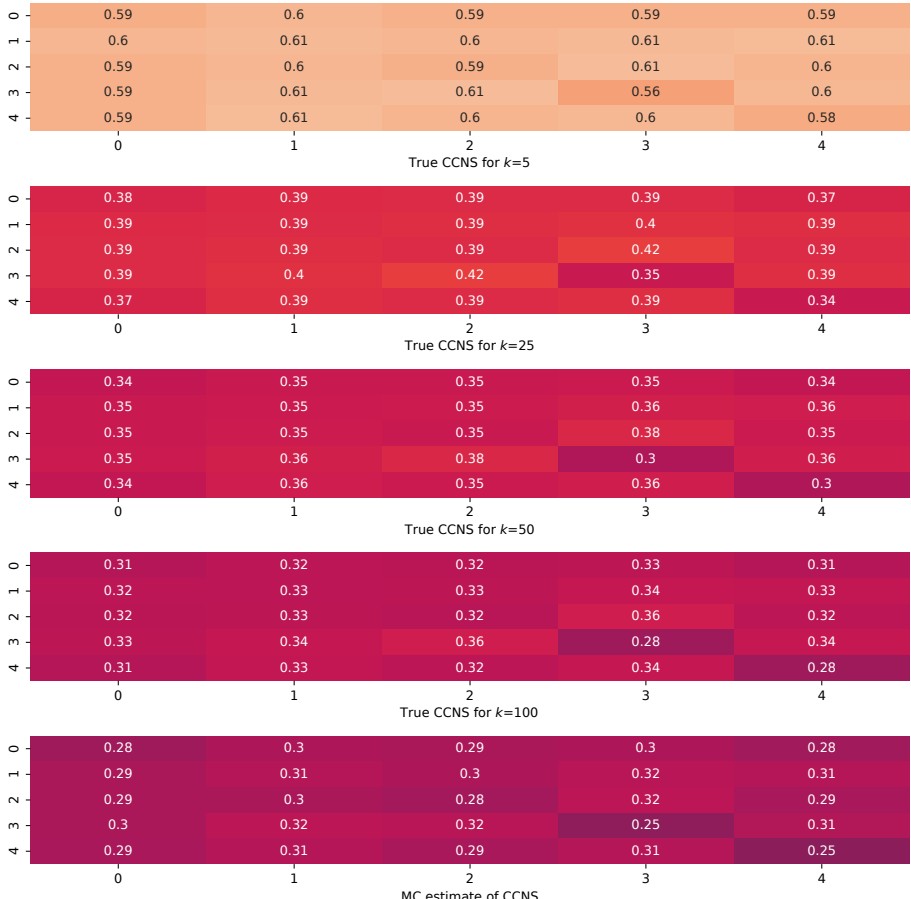

Figure 5: Comparison of the true CCNS for varying values of $k$ and the MC approximation according to our theory for a random initialization of a 5-class data model.

## A.6 Best Configurations and Ablation Studies

In the figure below, we show which are the best $k$ and $\varepsilon$ values picked by the models trained for the *qualitative* experiments.

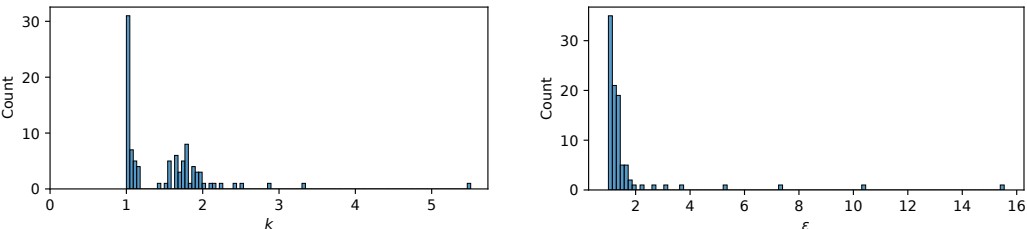

Figure 6: We show the histograms of the learned $k$ and $\varepsilon$ for the different configurations of Table 4. It appears that lower values of both parameters are preferred by the trained models.

**Analysis of Chosen Hyper-parameters** For completeness, in the following we display the best $k$, number of layers, and number of hidden units chosen by the different families of models on each dataset of Table 1. We recall that for each dataset and model there are 10 best configurations due to the use of a 10-fold cross validation for risk assessment [22]. We did not find any significant pattern except for sDGN preferring lower values of $k$ and the DGNs often choosing at least 2 convolutional layers.

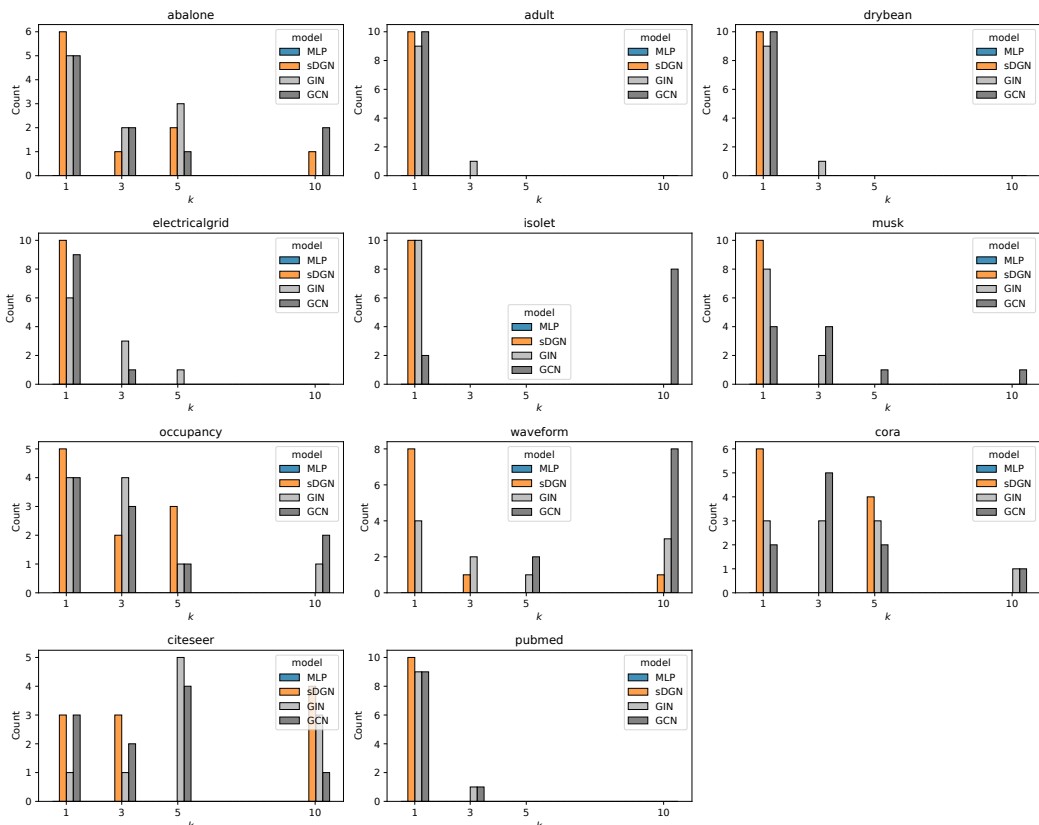

Figure 7: Histograms of the $k$ chosen by each model on the different datasets during a 10-fold cross validation for risk assessment.

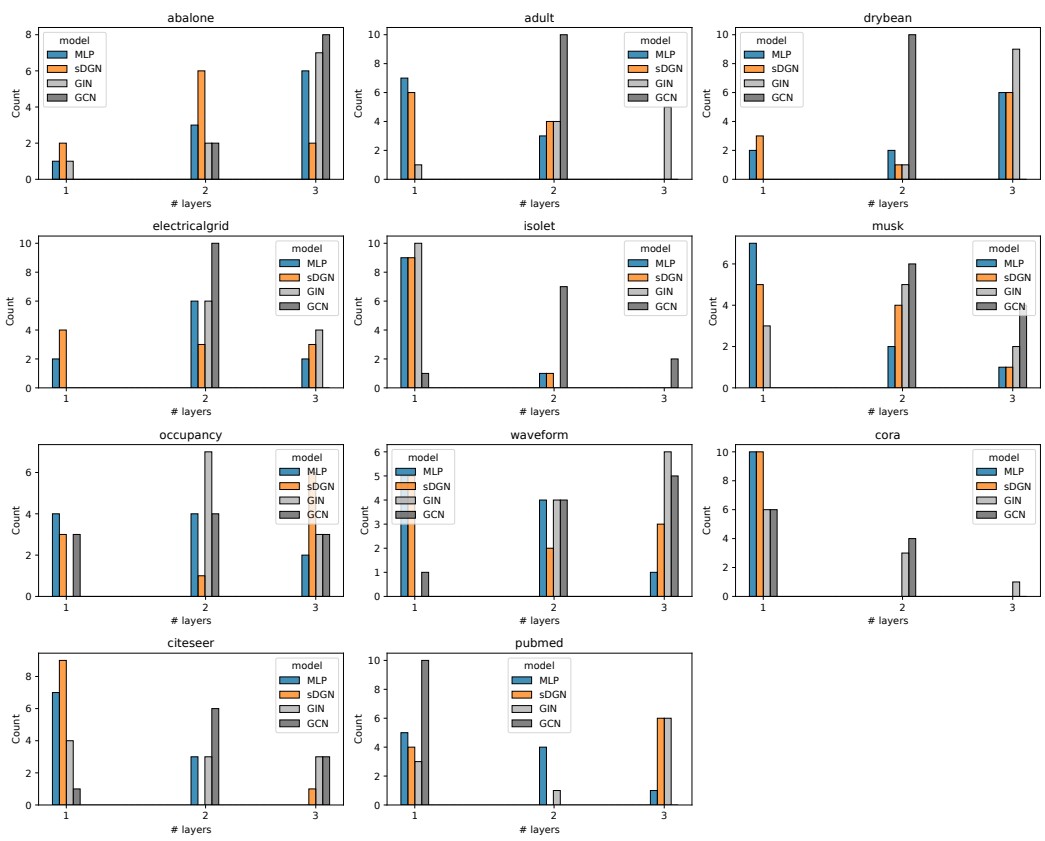

Figure 8: Histograms of the number of layers chosen by each model on the different datasets during a 10-fold cross validation for risk assessment.

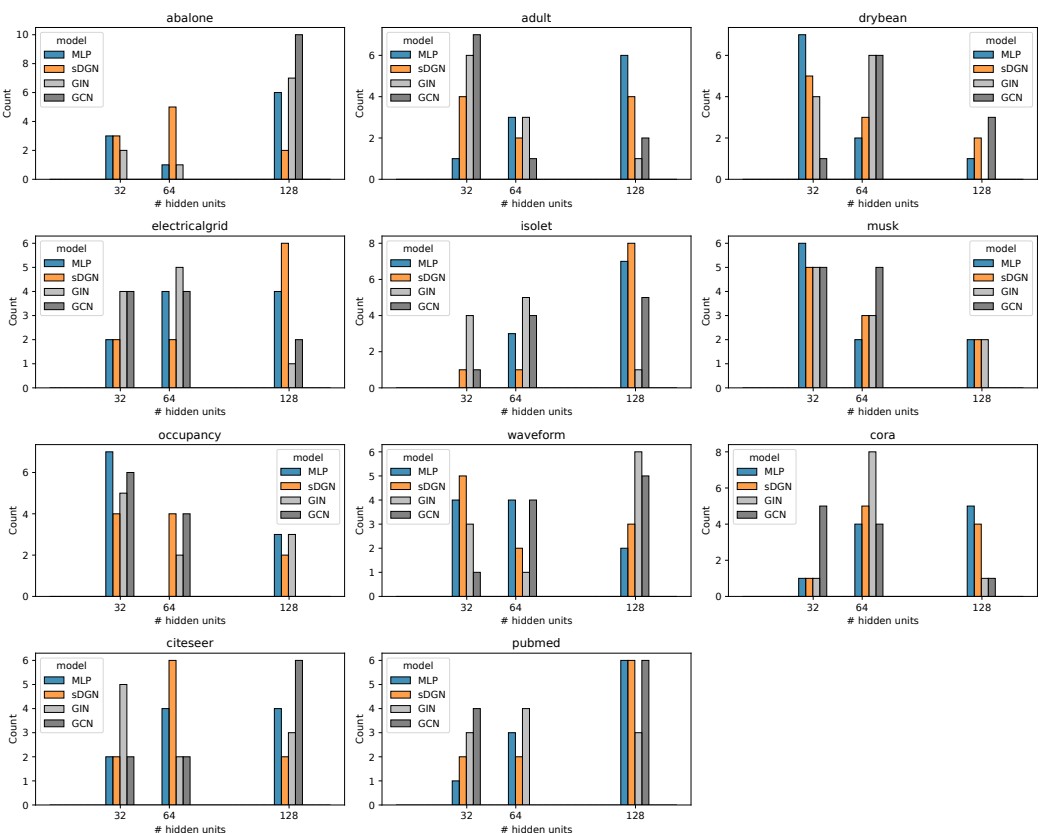

Figure 9: Histograms of the hidden units chosen by each model on the different datasets during a 10-fold cross validation for risk assessment.

**Ablation Study**    We now provide an ablation study of the validation performances of the models during the model selection phase (i.e., across all possible configurations). We show the ablations for the most important architectural choices of the hyper-parameters. Also in this case, a lower $k$ is associated with better performances, but there does not seem to be marked improvements when using more than 1 graph convolutional layer.

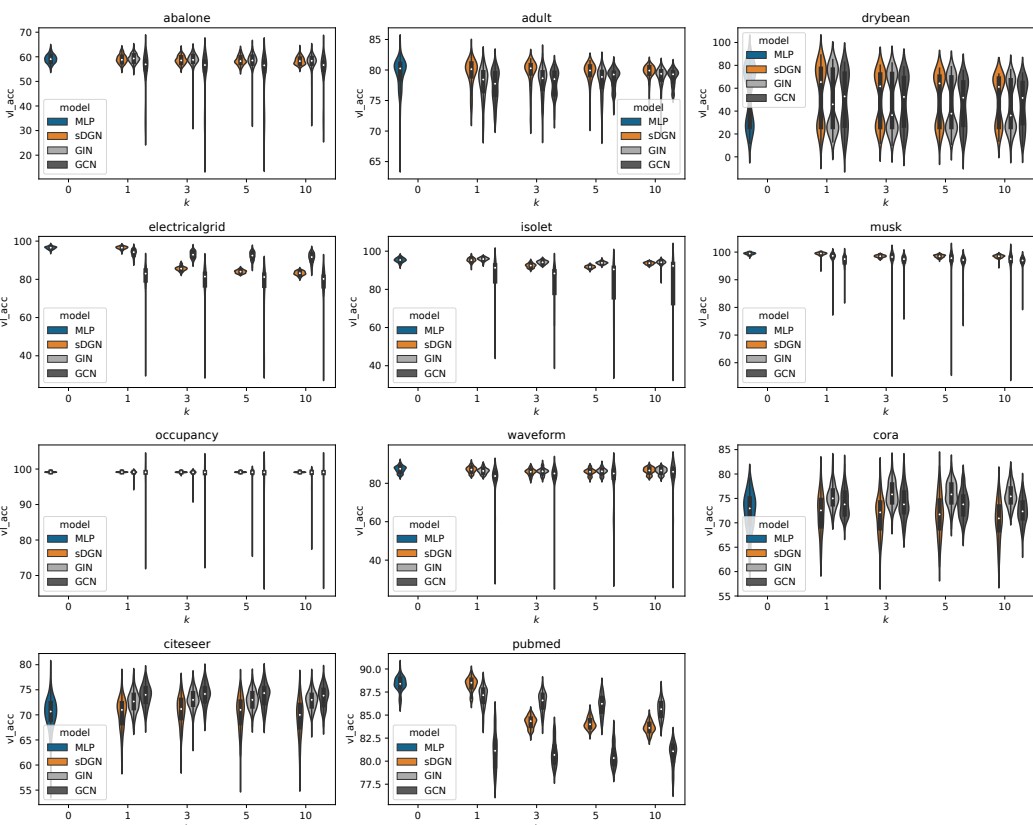

Figure 10: Ablation study for the $k$ chosen by each model on the different datasets. We report the validation performances computed during model selection for all configurations tried.

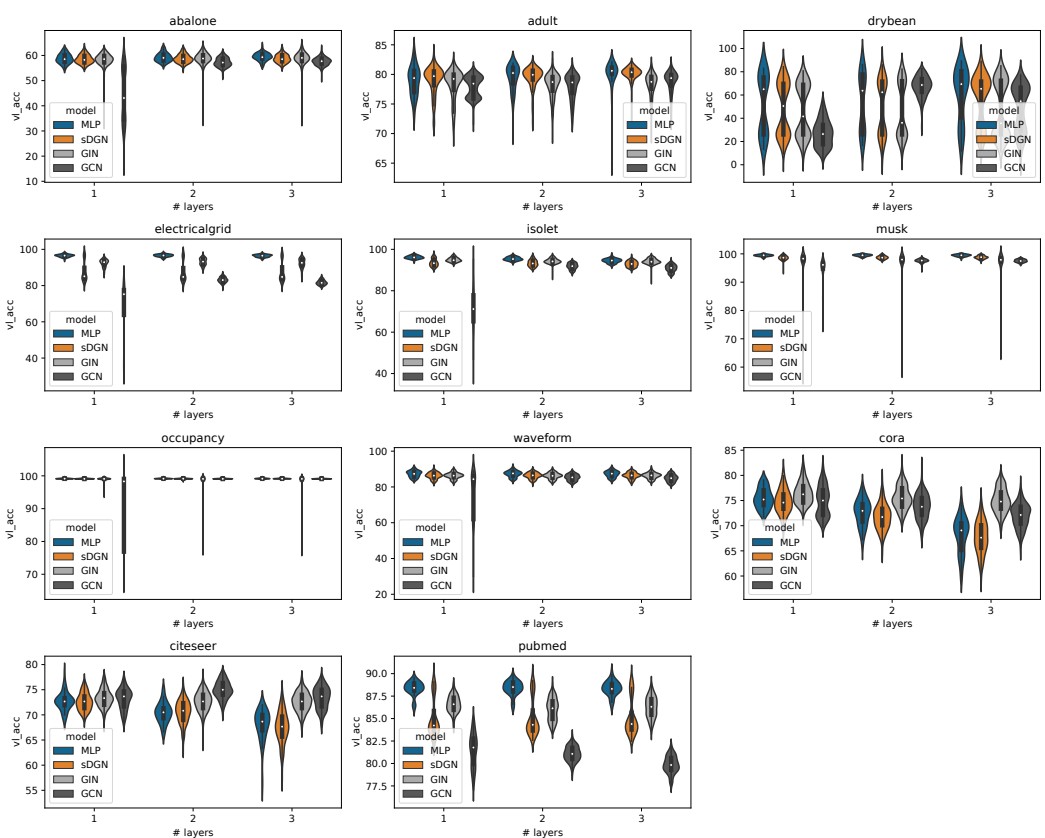

Figure 11: Ablation study for the number of layers chosen by each model on the different datasets. We report the validation performances computed during model selection for all configurations tried.

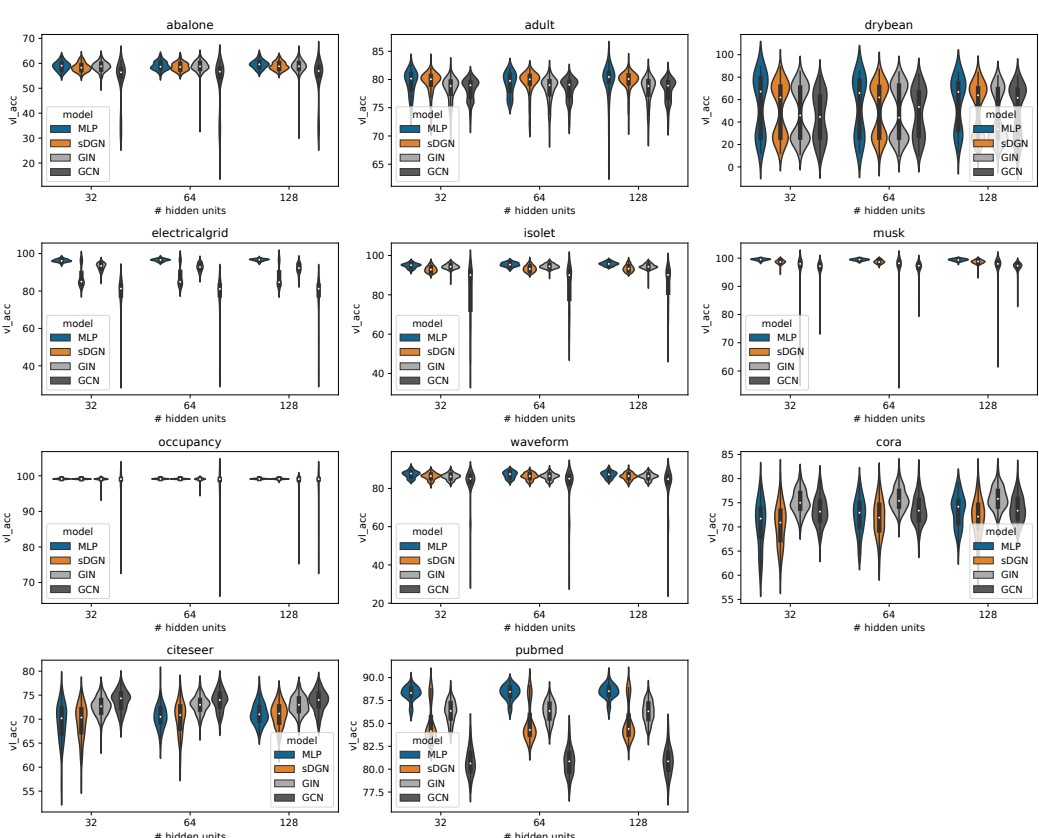

Figure 12: Ablation study for the hidden units chosen by each model on the different datasets. We report the validation performances computed during model selection for all configurations tried.

