# OpenReview forum: "On Class Distributions Induced by Nearest Neighbor Graphs for Node Classification of Tabular Data"
_NeurIPS.cc/2023/Conference — NeurIPS 2023 poster_

### Official Review · Reviewer_TGZz · 2023-06-16

**Soundness:** 2 fair
**Presentation:** 3 good
**Contribution:** 2 fair
**Rating:** 5
**Confidence:** 4

**Summary:**

This paper studies the machine learning problems on tabular data and convert the problem into node classification. Then, they formally analyze the Cross-Class Neighborhood Similarity (CCNS) and validate the performance on benchmark datasets.

**Strengths:**

S1. The studied problem is very important.

S2. The paper offers theoretical analysis.

S3. The paper is well written and the idea is easy to follow.

**Weaknesses:**

W1. More baselines are needed, such as other SOTA solutions in long-tail node classification.

W2. Neighbors’ sampling for balanced classes seems to be less novel [1]. The technical contribution is weak.

W3. Introducing graph data for tabular data has been studied a lot [2].

[1] Implicit Class-Conditioned Domain Alignment for Unsupervised Domain Adaptation, ICML 20

[2] TabGNN: Multiplex graph neural network for tabular data prediction.

**Questions:**

See weakness.

**Limitations:**

See weakness.

---

> ### Author Rebuttal · Authors · 2023-08-05
>
> We thank the reviewer for recognizing that the problem we study is important, and that we managed to present the theoretical ideas in a clear form.
>
> We address the questions/concerns of the reviewer below.
>
> **W1:** Long-tail node classification baselines deal, depending on its definition, with problem of power-law distributed degrees [*] or variable sized graphs [**]. In our experiments, each dataset correspond to a single graph where all nodes have exactly $k$ neighbors, so the inductive bias of the baselines suggested by the reviewer would not be advantageous in this context.
>
> To further clarify, the goal of the experimental approach is to show that k-NN approaches are not a panacea that always improves the performances on tabular data. If creating a k-NN graph was really helpful for the task, then we would observe improvements already with "more basic" methods like GCN and GIN.
>
> [*] Long-tailed graph neural networks via graph structure learning
> for node classification, Applied Intelligence, 2023.
>
> [**] On Size-Oriented Long-Tailed Graph Classification of Graph Neural Networks, WWW, 2022.
>
>
> **W2:** We are not entirely sure there is a connection between the referenced paper about unsupervised domain adaptation and what we are currently proposing. Also, our paper is not about neighbor’s sampling for balanced classes.
>
> We would appreciate if the reviewer could clarify this aspect.
>
> **W3:** We thank the reviewer for suggesting an additional reference to be included in the paper. There are many works, like the ones we mention in Section 2, that generate a graph for tabular data and try to empirically evaluate the performance of graph machine learning methods. However, **none** of these studies approach the problem from a theoretical perspective, and therefore the conclusions these studies draw remain limited to specific datasets.
>
> The novelty of our work mainly lies in a theoretical analyses that suggest how creating nearest neighbor graphs, and in particular k-NN graphs, might not be helpful to improve the performances in a fully supervised setting. The experimental part is only meant to bring additional empirical evidence supported by the theoretical results.
>
> In summary, while the introduction of graph for tabular data has been already studied, it is still unclear why and if it makes sense to do it in a fully supervised setup. Our work is the first theoretical study in this direction; we hope to have clarified this aspect.
>
> --
>
> We hope to have adequately addressed the points raised by the reviewer, but please let us know if further clarifications are needed.

---

> > ### Comment · Reviewer_TGZz · 2023-08-18
> > **Thanks for your response.**
> >
> > Thanks for your response. I will raise the score to 5.

---

### Official Review · Reviewer_MyQ5 · 2023-06-29

**Soundness:** 2 fair
**Presentation:** 2 fair
**Contribution:** 2 fair
**Rating:** 5
**Confidence:** 5

**Summary:**

The paper studies quantitative measures to evaluate the usefulness of k-NN graphs when performing classification in datasets that do not originally come with a known graph topology. The authors argue that the use of k-NN graphs is not useful.

**Strengths:**

A study of applying k-NN to construct a graph and then using standard GNN models for node classification. This is compared to using the baseline MLP that does not make use of any neighboring node information.

**Weaknesses:**

1. The paper is very heavy on assumptions, especially using the use of Gaussian models to derive certain results. There is no empirical evidence that such models hold in the datasets used.

2. The numerical experiments are not convincing enough. Maybe I misunderstood how the experiments are conducted. A major concern is how the k-NN graphs are constructed and how the GNN models are applied.

**Questions:**

1. In the definition (1), q_c does not seem to depend on the graph structure, unlike the original definition in [39]. How do you incorporate the notion of "neighborhood" here?
In fact, the whole development in Section 3.1 does not need to assume that we are performing node classification in a graph. Only in (9) do we use the notion of a node's neighborhood.

1. What is the purpose of the derivations in Prop 3.3 - 3.7? These are never used anywhere else in the paper or experiments (except for a very simple synthetic example). While I understand that the authors "leave these ideas to future investigations", there should be empirical studies to demonstrate that the results are of interest or useful.

1. The distribution p(\omega | c) in (3) has not been stated. Looking through the proofs and Prop 3.3, it seems to be assumed to be a Gaussian. Is this reasonable and is there empirical evidence for it?

1. The definition of SED in Prop 3.8 was never properly given. One has to look at the proof to realize the measure used is Lebesgue measure. Why is that when p(\omega | c) above is assumed to be Gaussian?

1. In the experiments, what distance metric is assumed to construct the k-NN graphs? The performance of different GNN models depends very much on the graph topology. Have different metrics been considered and do the authors' conclusions hold regardless of the metric?

**Limitations:**

Theoretical assumptions have not been comprehensively tested.

---

> ### Author Rebuttal · Authors · 2023-08-05
>
> We thank the reviewer for providing constructive feedback and the opportunity to clarify unclear points.
>
> **W1**: As we better discuss in our answers below, we rely on Gaussian **mixtures** that are known to be universal approximators. In addition, despite not covering all use-cases, assumption 2 is often used in the literature because they are particularly attractive from a theoretical perspective [*].
>
> In our opinion, assumptions 1 and 3 are quite reasonable considering the flexibility of Gaussian mixtures and the nearest neighbors construction (line 148) in the (implicitly Euclidean) message passing mechanism. In addition, we tested the non-linear assumption of Section 3.2 empirically, and we showed that it does not affect the analyses of Section 3.2.
>
> [*] Sutton, Charles, and Andrew McCallum. "An introduction to conditional random fields." Foundations and Trends in Machine Learning 4.4 (2012): 267-373.
>
> **W2**: We construct the experiment as follows:
> - Every dataset is a set of $n$ samples to be classified. For the GNN models, we construct a single k-NN graph of size $n$ using the Euclidean distance between attributes (see also our answer to the question below).
> - A GNN is trained on top of this graph to carry out the original classification problem as a node classification problem
>
> Our empirical setting follows the rigorous one of [19], trying many configurations for all baselines. Please let us know if we missed something we need to better clarify about this.
>
> **Q:** In the definition (1), q_c does not seem to depend on the graph structure, unlike the original definition in [39]. How do you incorporate the notion of "neighborhood" here?
>
> **A:** In nearest neighbor graphs, we can define a maximum neighboring distance $\epsilon$ from each node/sample. Hence, we incorporate the notion of neighborhood through the use of hyper-cubes (lines 150-152), which represent the set of all possible points (that is, neighbors) around the hyper-cube’s center (that is, the sample of interest).
>
> **Q + Limitations:** What is the purpose of the derivations in Prop 3.3 - 3.7? ... **(partially omitted for space limitations)**
>
> **A:** Improving the theoretical understanding of a phenomenon can influence the discovery of new practical methods, but not all theoretical insights have an easy and straightforward application. The theoretical developments of Section 3.1 are the first steps of a longer-term research plan on how to build *useful* synthetic graphs from tabular data, and to the best of our knowledge we are the first to do this from a theoretical perspective.
>
> The practical use of these results is left to future works simply because we would require an additional paper to first create an algorithm and then test it properly and convincingly.
>
> We hope this clarifies the long-term impact of our research and its relevance for future methods. We thank the reviewer for challenging us so that we can better improve the motivation for our studies in the paper.
>
> **Q:** The distribution p(\omega | c) in (3) has not been stated. Looking through the proofs and Prop 3.3, it seems to be assumed to be a Gaussian. Is this reasonable and is there empirical evidence for it?
>
> **A:** In the graphical model of Figure 1 (left), the distribution $p(x|c)$ (or equivalently $p(\omega | c)$) is a **mixture of Gaussians** conditioned on the class $c$. It is computed by marginalizing over the possible values of the variable $M$: $p(x|c) = \sum_m p(x|m,c)p(m|c)$. As an additional remark, mixtures of Gaussians are known to be universal approximators of distributions.
>
> We will fix the paper accordingly.
>
> **Q:** The definition of SED in Prop 3.8 was never properly given. One has to look at the proof to realize the measure used is Lebesgue measure. Why is that when p(\omega | c) above is assumed to be Gaussian?
>
> **A:** Thank you for spotting this, we will add the SED definition in the main paper at line 248. As discussed in our previous answer, $p(\omega | c)$ is a mixture of Gaussians and the SED is the only distance for which there exists an exact, rather than approximated, close-form equation for Gaussian mixtures [29].
>
> **Q:** In the experiments, what distance metric is assumed to construct the k-NN graphs? The performance of different GNN models depends very much on the graph topology. Have different metrics been considered and do the authors' conclusions hold regardless of the metric?
>
> **A:** We thank the reviewer for the insightful suggestion.  All related works (with the exception of [50] that uses the cosine similarity) rely on the Euclidean distance for two main reasons:
> - GNNs models typically work in the Euclidean space when aggregating neighbors, hence the homophily assumption can therefore be imposed - to a certain extent - only by computing nearest neighbors in terms of Euclidean distance
> - Euclidean distance has an easy interpretation in terms of “how similar are the attributes of adjacent nodes”
>
> Our analysis is set in this context, and we will clarify this in the paper. In addition, we considered the reviewer’s suggestion and ran additional experiments by using the cosine similarity as in [50] to construct the k-NN graphs. As shown in the PDF in the general response to reviewers, in general we observe generally worse performances of the GNN methods using this alternative metric, for instance on Adult and eGrid.
>
> This makes sense because, in addition to the above considerations, the use of cosine similarity is not “expressive” as the Euclidean distance as it ignores the magnitude of the vectors.
>
> --
>
> We hope to have adequately addressed the points raised by the reviewer. as we understand their importance in the final reviewer’s decision especially as regards the need for assumptions in a theoretical work. Thank you again for the opportunity to improve these aspects and we remain available should further clarifications be necessary.

---

> > ### Comment · Reviewer_MyQ5 · 2023-08-12
> >
> > Thank you for your clarifications. I appreciate the efforts by the authors to address the reviewers' comments. I just have two additional remarks.
> >
> > 1. Gaussian mixtures are universal approximators only for distributions with densities and in the sense of the weak* topology. They cannot approximate distributions with that have point masses. Moreover, even for a distribution with density, one will not know how many mixture components to use without prior information. These are reasons why you do not see much more widespread use of Gaussian mixtures in applications.
> >
> > 2. I agree that theoretical understanding is very important. Therefore, I believe that this would have been a much better paper if it focuses more deeply on the theoretical development with numerical experiments designed to test this theoretical understanding. For one, I appreciate the presented qualitative results more.

---

> > > ### Author Response · Authors · 2023-08-12
> > > **Response by Authors**
> > >
> > > We sincerely thank the reviewer for engaging in the discussion.
> > >
> > > - We agree with the clarification: this is indeed what we (partially) wrote at lines 136-137. We should have been more careful in our rebuttal and we will further clarify this point in the paper.
> > >
> > > - In Appendix A.4, Figure 4, we are additionally using the results of Prop 3.3-3.7 to approximate the CCNS of a synthetic dataset for varying values of $k$.
> > >
> > >   Also, the empirical results are what motivated us to work on the theoretical analysis, and a rigorous benchmarking in the fully-supervised setting is also lacking in the literature and needed to set a baseline. The rationale behind the structuring of the paper was that presenting a mixed set of results could have been positively received by a broader community.
> > >
> > >
> > > As we value the reviewer’s feedback, we would like to do everything we can to improve the quality of the paper at this stage and possibly raise the score. Is there any additional analysis that we could provide to the reviewer to achieve this?
> > >
> > > Thank you again for your time. We look forward to continuing the discussion.

---

> > > > ### Author Response · Authors · 2023-08-14
> > > > **Additional numerical experiments**
> > > >
> > > > Dear Reviewer,
> > > >
> > > > Below, we provide two additional numerical analyses that further investigate the behavior of the theoretical results of Section 3.1. We cannot update the pdf nor send links to images, so we have to resort to tables for this rebuttal. However, we will update the paper with the figures, which are easier to interpret and cover a broader spectrum of values.
> > > >
> > > > **Analysis of behavior of Theorem 3.5 (depends on Prop 3.6)**
> > > >
> > > > Here, we took the example of Figure 1 and computed the $2\times 2$ matrix $M_c(c', \varepsilon)$ for varying values of $\varepsilon$, to show how it behaves when the hypercube is made increasingly larger. As we can see from the table, when  $\varepsilon$ is very small the probability that a neighbor is of the same class is very high for both classes (but not symmetric), since there is relatively small overlap between the green (c=0) and blue (c=1) distributions in the Figure. As we increase the side length of the hypercube (or equivalently, we consider more and more nearest neighbors), the each row of the matrix converges to $[0.5, 0.5]$, which corresponds to the prior $p(c)$ that we set for this example. This result is in agreement with the second limit of Proposition $3.7$.
> > > >
> > > > |          |        | $c=0$ | $c=1$ |
> > > > |----------|--------|-------|-------|
> > > > | eps=0.01 | $c'=0$ | 0.87  | 0.13  |
> > > > |          | $c'=1$ | 0.10  | 0.90  |
> > > > |          |        | $c=0$ | $c=1$ |
> > > > | eps=1    | $c'=0$ | 0.85  | 0.15  |
> > > > |          | $c'=1$ | 0.14  | 0.86  |
> > > > |          |        | $c=0$ | $c=1$ |
> > > > | eps=3    | $c'=0$ | 0.72  | 0.28  |
> > > > |          | $c'=1$ | 0.27  | 0.73  |
> > > > |          |        | $c=0$ | $c=1$ |
> > > > | eps=5    | $c'=0$ | 0.56  | 0.44  |
> > > > |          | $c'=1$ | 0.43  | 0.57  |
> > > > |          |        | $c=0$ | $c=1$ |
> > > > | eps=10   | $c'=0$ | 0.5   | 0.5   |
> > > > |          | $c'=1$ | 0.5   | 0.5   |
> > > >
> > > >
> > > > **Analysis of behavior of Proposition 3.7 for varying $\varepsilon$ and SED(X_0, X_1)**
> > > >
> > > > We now analyze the left limit of Proposition 3.7 for the example in Figure 1, where $D=1$ and hence the assumptions are satisfied. We translated the green curve to the right by different values (from 0 to 4) and computed the $SED(X_0,X_1)$ between the green and blue curves at each step, together with the matrix $lim_{\varepsilon \rightarrow 0}M_c(c', \varepsilon)$. The goal is to show the relation between the two quantities. We observe that the behavior of $lim_{\varepsilon \rightarrow 0}M_c(c', \varepsilon)$ is non-trivial when the two curves overlap (specifically for $SED(X_0,X_1) < 5$). In particular, the greater the overlap the higher the entropy for the rows of the matrix, which makes sense since it becomes harder to distinguish samples of different classes. On the other hand, the behavior becomes trivial when the curves are distant enough from each other: there, in the limit, every node will only have neighbors of the same class.
> > > >
> > > > |                  |        | $c=0$ | $c=1$ |
> > > > |------------------|--------|-------|-------|
> > > > | translation = 0. | $c'=0$ | 1     | 0     |
> > > > | SED = 7.85       | $c'=1$ | 0     | 1     |
> > > > |                  |        | $c=0$ | $c=1$ |
> > > > | translation = 1. | $c'=0$ | 0.68  | 0.32  |
> > > > | SED = 4.47       | $c'=1$ | 0.26  | 0.74  |
> > > > |                  |        | $c=0$ | $c=1$ |
> > > > | translation = 2. | $c'=0$ | 0.60  | 0.40  |
> > > > | SED = 0.99       | $c'=1$ | 0.33  | 0.64  |
> > > > |                  |        | $c=0$ | $c=1$ |
> > > > | translation = 3. | $c'=0$ | 0.62  | 0.38  |
> > > > | SED = 2.73       | $c'=1$ | 0.31  | 0.69  |
> > > > |                  |        | $c=0$ | $c=1$ |
> > > > | translation = 4. | $c'=0$ | 1     | 0     |
> > > > | SED = 9.23       | $c'=1$ | 0     | 1     |
> > > >
> > > > --
> > > >
> > > > As mentioned, we will include these tables as figures in the paper and complement them with the above discussion. There should be enough room to include these in the camera-ready version. We thank again the reviewer for helping us to improve the quality of our paper. We remain available for should further questions or analyses be needed.

---

> > > > > ### Comment · Reviewer_MyQ5 · 2023-08-17
> > > > >
> > > > > Thank you for the additional results, which support the theoretical analysis. I will raise my score.

---

> > > > > > ### Author Response · Authors · 2023-08-17
> > > > > > **Response**
> > > > > >
> > > > > > Thank you very much for your help and constructive feedback.

---

### Official Review · Reviewer_Ysph · 2023-07-05

**Soundness:** 2 fair
**Presentation:** 3 good
**Contribution:** 2 fair
**Rating:** 6
**Confidence:** 2

**Summary:**

This paper shows that using a k-NN graph in tabular data classification is not beneficial compared to MLP. Specifically, based on Cross-Class Neighborhood Similarity, the authors theoretically induce the analytical forms of the squared error distance (SED) of two approaches. In the experiment, they demonstrate that the SED of the method using a k-NN graph is smaller than the SED of the method using attributes directly. Furthermore, in fully-supervised setting, they show that MLP exhibits comparable performance with methods using k-NN graphs.

**Strengths:**

- They propose a theoretical framework to analyze the method using a k-NN graph.
- They show that the SED for using a k-NN graph is smaller than the SED for using attributes directly in synthetic data.
- They demonstrate that MLP exhibits comparable performance with the approaches using a k-NN graph in fully-supervised setting.

**Weaknesses:**

- [W1] As the authors mentioned, the assumptions seem strong. Especially, for assumption 1, it would not hold that attributes are independent in the real-world data. Also, they fix GNN architecture as GCN in Equation 9, but there exists many effective GNNs.
- [W2] I understand that SED can be related with performance. However, it seems not enough to exploit the statement that the methods with large SED would show better performance than the methods with small SED.
- [W3] As far as I understand, the proposed framework gives only the analytic forms of SED and the empirical evidence for SED difference is also suggested in the synthetic setting. It seems not enough to assert that using a k-NN graph offers no benefits compared to a structure-agnostic baseline.
- [W4] The analysis about the analytic forms does not provided in the main paper.

**Questions:**

- Following [W1], could you provide the correlation between attributes in the real-world datasets?
- Following [W2], would you provide eivdences about the relation between SED and classification performance?
- Following [W3], could you provide more evidences that the SED for the methods using a k-NN graph is smaller than the SED of the methods using attributes directly?
- Following [W4], would you provide the detailed description about analytic forms? Also, I'm curious about the conditions when SED of methods using a k-NN graph gets large based on the analytic forms?

**Limitations:**

As the authors mentioned, the assumptions seem strong [W1].

---

> ### Author Rebuttal · Authors · 2023-08-05
>
> We sincerely thank the reviewer for providing constructive feedback. Below, we address the reviewer questions, each of which is related to a specific weakness.
>
> **Q:** Following [W1], could you provide the correlation between attributes in the real-world datasets?
>
> A:  Please have a look at the PDF in the general response to reviewers. We provide the Pearson correlation analysis for two datasets that have few attributes in the pdf attached to this submission. However, we would like to remind the reviewer that *correlation does not imply causation*, hence we cannot still draw conclusions about the form of the true data generating distribution.
>
> In our opinion, assumptions 1 and 3 are quite reasonable considering the flexibility of mixture models and the nearest neighbors construction (line 148). In addition, we tested the non-linear assumption of Section 3.2 empirically, and we showed that it does not affect the analyses of Section 3.2.
>
> Assumptions 2, instead, is widely used in the literature and make graphical models particularly attractive from a theoretical perspective [*], despite it is also known that they cannot cover all real-world cases.
>
> [*] Sutton, Charles, and Andrew McCallum. "An introduction to conditional random fields." Foundations and Trends in Machine Learning 4.4 (2012): 267-373.
>
> **Q:** Following [W2], would you provide evidences about the relation between SED and classification performance?
>
> **A:** We can mention a few of the numerous works in the literature (see list below) where this relation, in the more general case of distance metrics and not just SED, is discussed and analyzed. Thank you for pointing this out, we will include a clarification with the references in the paper.
>
> - Oneto, et al. "Do we really need a new theory to understand over-parameterization?." Neurocomputing 543 (2023).
> - Biggio and Roli. "Wild patterns: Ten years after the rise of adversarial machine learning." Proceedings of the 2018 ACM SIGSAC Conference on Computer and Communications Security. 2018.
> - Elsayed, Gamaleldin, et al. "Large margin deep networks for classification." Advances in neural information processing systems 31 (2018).
> - Vapnik,. "Statistical learning theory." Wiley series on adaptive and learning systems for signal processing, communications and control (1998).
> - Oh, Il-Seok et al.,  "Analysis of class separation and combination of class-dependent features for handwriting recognition." IEEE Transactions on Pattern Analysis and Machine Intelligence 21.10 (1999).
>
> **Q:** Following [W3], could you provide more evidences that the SED for the methods using a k-NN graph is smaller than the SED of the methods using attributes directly?
>
> **A:** Because assumptions about data are inevitable, it is very hard to prove that a k-NN graph *never* helps. However, below we provide more theoretically inspired evidence as to why this should be always the case -- at least under our assumptions.
>
> In the proof of Prop 3.8 (appendix), we observe that $p(h|c)$ is a mixture of Gaussians that differs from $p(x|c)$ only in the higher variance of the individual Gaussian components. In other words, the distribution of $p(h|c)$ looks “flatter” than $p(x|c)$. At least in a one or two-dimensional space, flatter distributions $p(h|c)$ and $p(h|c’)$ should therefore increase the overlap (and therefore decrease the SED) compared to the SED in the original space. This intuition, despite its simplicity, is unfortunately hard to prove for technical reasons, for instance the dependence of the distance value on the specific parameters $\mu$ and $\Sigma$.
>
> We hope this addresses the reviewer's question, but please let us know if there is something else specific we should clarify.
>
> **Q:** Following [W4], would you provide the detailed description about analytic forms? Also, I'm curious about the conditions when SED of methods using a k-NN graph gets large based on the analytic forms?
>
> **A:** Due to space reasons we could not include the analysis in the main paper and had to leave our considerations in the appendix (lines 659-672). We might be able to include those, however, in the camera ready version.
>
> This is a great question and related to our answer above. Without making assumptions over the values of the $\mu$ and $\Sigma$ of the true data generating distributions, one cannot easily see when the SED gets large. However, since the SED of the embedding space seems to be (empirically) always smaller than the SED of the original space and closely related in shape (theoretically), we can argue that the former should get larger as the latter increases. This happens when the class-specific distributions in the original space get farther away from each other, which makes the use of a k-NN graph even less useful because different classes become more separable.
>
> --
>
> We thank the reviewer for the questions that touch profound points of our work. We hope to have provided satisfactory answers to the reviewer’s curiosity and to have addressed the doubts. We remain available for continuing a constructive discussion should clarifications be needed.

---

> > ### Comment · Reviewer_Ysph · 2023-08-12
> > **Change my score to weak accept**
> >
> > I appreciate your detailed response. Most of my concerns are resolved. I change my score to weak accept in that the key message is significant and the logical reasoning seems correct.

---

> > > ### Author Response · Authors · 2023-08-17
> > > **Response**
> > >
> > > We thank the reviewer for appreciating our efforts and for helping us to further improve the quality of our paper.

---

### Official Review · Reviewer_CVih · 2023-07-10

**Soundness:** 3 good
**Presentation:** 2 fair
**Contribution:** 3 good
**Rating:** 5
**Confidence:** 1

**Summary:**

This submission provides a theoretical framework to explore the benefits of Nearest Neighbor (NN) graphs in the absence of a predefined graph structure. The Cross-Class Neighborhood Similarity (CCNS) is formally analyzed to evaluate the usefulness of structures within nearest neighbor graphs. The study also investigates the class separability induced by deep graph networks on a k-NN graph. Quantitative experiments conducted under full supervision demonstrate that employing a k-NN graph provides no advantages compared to a structure-agnostic baseline. Qualitative analyses indicate that the framework effectively estimates CCNS and suggests that k-NN graphs are not useful for such classification tasks. This highlights the need to explore alternative graph construction techniques.

**Strengths:**

1. The theoretical analysis is interesting.

2. The motivation is clear.

**Weaknesses:**

To be honest, I am not familiar with this topic. I can only provide high-level comments as follows,

1. Equation (2) is not clear. For example, how to derive the equivalent between two absolute values?

2. Figure 2 left is not very easy to be understood. For example, what are the colors mean, and what is the meaning of each line?

**Questions:**

N/A

**Limitations:**

The authors have discussed the limitations.

---

> ### Author Rebuttal · Authors · 2023-08-05
>
> We thank the reviewer for taking the time to read and assess our work. Below we provide an answer to the questions posed.
>
> **Q1:** Equation (2) is not clear. For example, how to derive the equivalent between two absolute values?
>
>
> **A:** The equivalence of Eq. 2 follows from the linearity of expectations and from the fact that $q_c(x)$ does not depend on $p(x|c’)$ and vice versa ($q_c’(x),p(x|c)$), so the two resulting expectations further simplify:
>
> $\\mathbb{E}\_{p(x|c)p(x'|c')}[q\_c(x) - q\_{c'}(x')] = \\mathbb{E}\_{p(x|c)\\cancel{p(x'|c')}}[q\_c(x)] -  \\mathbb{E}\_{\\cancel{p(x|c)}p(x'|c')}[q\_{c'}(x')] = \\mathbb{E}\_{p(x|c)}[q\_c(x)] -  \\mathbb{E}\_{p(x'|c')}[q\_{c'}(x')] $
>
> Thanks for mentioning this, we will add it to the paper.
>
> **Q2:** Figure 2 left is not very easy to be understood. For example, what are the colors mean, and what is the meaning of each line?
>
> **A:** With Figure 2, we want to empirically test whether there exist tabular datasets (following our assumptions) where a nearest-neighbor structure can help to better discriminate the samples. Each curve/color corresponds to a different configuration of hyper-parameters (Table 4 appendix) for the data generating distribution of the dataset.
>
> The value on the y-curve indicates if the embedding space constructed using $k$-NN graphs and DGNs is more separable than the original space. To be more separable, the value should exceed 1, but we observe that the lines never cross this value and reach an asymptote as we increase $k$. Therefore, $k$-NN graphs do not seem to help when the data generating distribution behaves according to our theory.
>
> --
>
> We hope to have adequately addressed the concerns of the reviewer. If not, please let us know and we will do our best to further clarify these points.

---

> > ### Comment · Reviewer_CVih · 2023-08-12
> > **Response**
> >
> > Thank you for your response. My concerns are solved.
> >
> > Since I am not familiar with this topic, I will follow other reviewer's final judgment

---

> > > ### Author Response · Authors · 2023-08-21
> > > **Response to Reviewer CVih**
> > >
> > > Thank you for your message; we understand the reviewer's position.
> > >
> > > On a separate note, since the rebuttal is closing soon, we were hoping you could still consider adjusting the score as we have addressed all the reviewer's concerns. Thank you again for your time.

---

### Official Review · Reviewer_Cqgk · 2023-07-24

**Soundness:** 2 fair
**Presentation:** 3 good
**Contribution:** 3 good
**Rating:** 6
**Confidence:** 3

**Summary:**

The main contributions of this paper are as follows:

1. The authors of the paper study Cross-Class Neighborhood Similarity (CCNS) for nearest neighbor graphs and provide its first lower bound.
2. They apply a Deep Graph Network (DGN) to an artificial k-NN graph and study how it affect the class separability of the input data.
3. They carry out a robust comparison between structure-agnostic and structure-aware baselines on a set of 11 datasets, and their results are in agreement with their theoretical results.
4. Qualitative analyses based on the theory further suggest that using the k-NN graph might not be advisable.

Overall, the authors introduce a theoretical framework to understand the benefits of Nearest Neighbor (NN) graphs when a graph structure is missing and provide both theoretical and empirical evidence to support their conclusions. They try to raise awareness about the potential limitations of k-NN graphs for node classification tasks and advocates for the study of alternative graph construction techniques.

**Strengths:**

1. The authors introduce a theoretical framework to understand the benefits of Nearest Neighbor (NN) graphs when a graph structure is missing.
2. They provide a formal analysis of the Cross-Class Neighborhood Similarity (CCNS) in the context of nearest neighbor graphs and provide an analytical computation of the CCNS for Nearest Neighbor Graphs.
3. They carry out a robust comparison between structure-agnostic and structure-aware baselines on a set of 11 datasets that is in agreement with their theoretical results.

Overall, this paper provides both theoretical and empirical evidence to support its conclusions and makes a contribution to the understanding of the benefits and limitations of using k-NN graphs for node classification tasks.

**Weaknesses:**

As the authors mentioned in the paper, they assume that the attributes are conditionally independent when conditioned on a specific mixture and class. These assumptions may not hold for all datasets and could limit the generalizability of their results.

Additionally, the authors’ analysis is focused on the fully-supervised setting, where all training labels are available. It is not clear how their conclusions would apply to other settings, such as the semi-supervised setting where only a subset of training labels is available.

Overall, while this paper provides valuable insights into the benefits and limitations of using k-NN graphs for node classification tasks, its conclusions are based on specific assumptions and conditions and may not apply to all situations.


**Questions:**

1. How does your work relate to other dataset formats more than tabular data?

2. How would your conclusions apply to other settings, such as the semi-supervised setting where only a subset of training labels is available?

---

> ### Author Rebuttal · Authors · 2023-08-05
>
> We thank the reviewer for recognizing the merits and strengths of our work. Below, we will address the comments and questions that have been raised.
>
> **On the independence assumption**
>
> The conditional independence assumption has been widely used in the graphical models literature because it is particularly attractive from a theoretical perspective [*], despite it cannot cover all real-world cases.
>
> We also want to remark that this assumption is only needed in Section 3.1 to approximate the CCNS, and it is not required in Section 3.2 which deals with the more practical problem of class separability induced by DGNs.
>
> Assumptions 1 and 3 are still quite reasonable considering the flexibility of mixture models and the nearest neighbors construction (line 148). In addition, we tested the non-linear assumption of Section 3.2 empirically, and we showed that it does not affect the analyses of Section 3.2.
>
> Overall, the results we have obtained help us better understand the observed behaviour of models under speficic conditions, but they also pave the way for other researchers to extend these results to more general scenarios, which is usually the case for theoretical works.
>
> [*] Sutton, Charles, and Andrew McCallum. "An introduction to conditional random fields." Foundations and Trends in Machine Learning 4.4 (2012): 267-373.
>
> **On the fully supervised setting**
>
> Our results do not apply to the semi-supervised setting: it has already been shown in the literature (lines 31-48) that a k–NN graph can help in the semi-supervised setting, possibly due to a regularizing effect that prevents overfitting of the few labels.
>
> For this reason, we study the only alternative scenario where the above conclusions have not been comprehensively tested yet, and our theory suggests an opposite result in the fully supervised context.
>
> **Q:** How does your work relate to other dataset formats more than tabular data?
>
> **A:** Our work is specific to tabular data tasks that are transformed into node classification tasks, as it is becoming a more and more common approach.
>
> **Q:** How would your conclusions apply to other settings, such as the semi-supervised setting where only a subset of training labels is available?
>
> **A:** please refer to our considerations above.
>
> --
>
> We hope to have adequately addressed the reviewer’s doubts. Please let us know if there is something we should further clarify.

---

> > ### Author Response · Authors · 2023-08-19
> >
> > Thank you again for your positive and valuable feedback.
> >
> > We hope our responses addressed your concerns. As the rebuttal is closing soon, we would appreciate if the reviewer could consider raising the score accordingly. We remain available for further requests of clarifications should they be needed. Your feedback is greatly appreciated.

---

### Official Review · Reviewer_VEGV · 2023-07-26

**Soundness:** 4 excellent
**Presentation:** 3 good
**Contribution:** 3 good
**Rating:** 7
**Confidence:** 2

**Summary:**

This paper proposes to investigate the practice of using nearest neighbour graphs build upon tabular data in order to classify samples (then relying on a node classification model). The authors note that this practice has been shown to be relevant in the semi-supervised case, but not with full supervision - and that it often exploits the common idea that homophily is a desired graph property for node classification models. However, recent works shows that the difference between the distributions of labels of neighbours for nodes of separate class (measured by the Cross-Class Neighborhood Similarity, CCNS) is what matters for their efficiency. This brings the authors to propose (1) a formal model for computing an approximation of these distributions under several assumptions (2) under the same assumptions, study the separability of classes induced by a simple node representation model relying on message passing on the NN-graph.
The authors then propose two sets of experiments to draw conclusions: (1) an study of the results of learning classification models directly on tabular data or on NN-graphs for 11 tasks (2) qualitative experiments relying on their theoretical work, verifying if, for a toy task of binary classification based on simple distributions built following their assumptions: (a) there exists of configuration where classes are more separable using a NN-graph structure (b) the quantities obtained via their approximation of CCNS concurr to the conclusion of (a).

**Strengths:**

- This paper is very well written, and easy to follow, despite the difficulty of the subject.
- This paper demonstrates experimentally that using nearest neighbour graphs build upon tabular data in order to classify samples does not help performance.
- This paper provides well-motivated theoretical arguments showing that a model built on message-passing for a k-NN graph will not improve class separability.
- Finally, this paper provides a theoretical framework (under assumptions) to build a tool, allowing to approximate the CCNS (an indicator of potential success of node classification models).

**Weaknesses:**

- While the paper clearly discusses the limitations brought by the assumptions it makes, it would be interesting and useful to explore how significant they are.
- The utility of the theoretical developments for the experimental demonstration is not made completely clear.

**Questions:**

- The purpose of the CCNS approximation proposed is a little unclear. Is it supposed to be a practical tool for testing the relevance of the k-NN graph for a classification task ?

**Limitations:**

- Limitations have been adequately addressed.

---

> ### Author Rebuttal · Authors · 2023-08-05
>
> We are glad that the reviewer found our work well presented and motivated. Below, we comment on the points raised by the reviewer, hoping that it will better clarify the scope of this work.
>
> **Q:** While the paper clearly discusses the limitations brought by the assumptions it makes, it would be interesting and useful to explore how significant they are.
>
> **A:** A challenge, common to all theoretical machine learning problems like this one, in verifying how impactful the assumptions are is that we do not have access to the true data generating distribution of real-world datasets.
>
> For instance, analyses that focus on variable correlations cannot ensure that there is a real causation between variables in the dataset, so the assumption of independence cannot be easily verified.
>
> However, we have at least empirically tested the impact of non-linearities on the results by comparing sDGN with GIN and GCN, and we have observed no influence on the performances. In addition, the use of a mixture of Gaussians in the data generating model allows us to represent any distribution with a sufficiently high number of mixtures, so it is not a limiting assumption.
>
> **Q:** The utility of the theoretical developments for the experimental demonstration is not made completely clear.
>
> **A:** Improving the theoretical understanding of a phenomenon can influence the discovery of new practical methods, but not all theoretical insights have an easy and straightforward application. The theoretical developments of Section 3.1 are the first steps of a longer-term research plan on how to build *useful* synthetic graphs from tabular data, and to the best of our knowledge we are the first to do this from a theoretical perspective.
>
> In contrast, the theoretical studies of Section 3.2 are more directly related to the evaluation of various methods under the k-NN graph construction, which is why we could provide a thorough empirical analysis on that end.
>
> --
>
> We hope to have addressed the doubts of the reviewer, but please let us know if we should further clarify some aspects of our work. Thank you again for the constructive feedback.

---

### Author Rebuttal · Authors · 2023-08-08

We sincerely thank all reviewers for their constructive feedback. We did our best to answer all questions and we look forward to a fruitful discussion.

Attached you can find a PDF with additional results to answer the reviewers' questions. Please note that the Table is partially complete due to running experiments, and we will do our best to complete it or to update you during the discussion period.

---

> ### Author Response · Authors · 2023-08-12
> **Update: Results**
>
> Dear reviewers,
>
> We have finalized the experiments of the Table in the pdf where we build k-NN graphs using the cosine similarity rather than Euclidean distance. All results are still on par of far worse than an MLP. Please refer to the answer to Reviewer MyQ5 for a discussion about why this might be the case.
>
> |           | MLP         | MLP        | sDGN        | sDGN       | GIN         | GIN        | GCN         | GCN        |
> |-----------|-------------|------------|-------------|------------|-------------|------------|-------------|------------|
> |           | F1          | ACC        | F1          | ACC        | F1          | ACC        | F1          | ACC        |
> | Abalone   | 0.55 (0.02) | 55.5 (1.5) | 0.55 (0.01) | 55.5 (0.9) | 0.55 (0.03) | 55.2 (1.8) | 0.55 (0.02) | 55.1 (1.9) |
> | Adult     | 0.70 (0.02) | 77.6 (1.7) | 0.64 (0.01) | 79.0 (0.6) | 0.69 (0.03) | 79.5 (0.6) | 0.53 (0.07) | 76.4 (0.6) |
> | DryBean   | 0.78 (0.04) | 77.4 (4.0) | 0.77 (0.06) | 77.0 (5.0) | 0.79 (0.02) | 78.2 (2.0) | 0.78 (0.06) | 78.2 (5.3) |
> | eGrid     | 0.95 (0.01) | 95.6 (0.6) | 0.84 (0.01) | 85.8 (0.9) | 0.93 (0.01) | 93.9 (0.8) | 0.81 (0.01) | 82.6 (1.4) |
> | Isolet    | 0.95 (0.01) | 95.4 (0.5) | 0.93 (0.01) | 92.7 (1.0) | 0.95 (0.01) | 94.8 (1.0) | 0.90 (0.05) | 90.6 (4.1) |
> | Musk      | 0.99 (0.01) | 99.3 (0.3) | 0.96 (0.01) | 98.0 (0.6) | 0.93 (0.04) | 96.2 (2.5) | 0.93 (0.04) | 97.1 (0.7) |
> | Occupancy | 0.99 (0.01) | 98.9 (0.4) | 0.99 (0.01) | 98.6 (0.3) | 0.99 (0.01) | 98.9 (0.3) | 0.95 (0.1)  | 97.6 (4.8) |
> | Waveform  | 0.85 (0.02) | 85.5 (1.7) | 0.84 (0.01) | 84.0 (1.4) | 0.84 (0.01) | 84.1 (1.6) | 0.82 (0.03) | 82.6 (2.3) |
> | Cora      | 0.71 (0.04) | 74.3 (2.4) | 0.71 (0.03) | 74.1 (2.7) | 0.72 (0.02) | 75.5 (1.8) | 0.72 (0.03) | 75.2 (2.3) |
> | Citeseer  | 0.69 (0.02) | 71.8 (1.9) | 0.69 (0.02) | 71.8 (1.8) | 0.70 (0.02) | 73.7 (1.3) | 0.69 (0.02) | 71.7 (2.4) |
> | Pubmed    | 0.87 (0.01) | 87.6 (0.7) | 0.84 (0.01) | 83.8 (0.7) | 0.86 (0.01) | 86.7 (0.8) | 0.84 (0.01) | 83.7 (0.6) |
>
> We hope the reviewers appreciate our effort and we remain available for further clarifications.

---

### Decision · Program_Chairs · 2023-09-21

**Decision:**

Accept (poster)

**Comment:**

In this paper the authors present a theoretical and empirical analysis for understanding the impact of k-Nearest Neighbor (NN) graphs on classification problems with tabular data, where one does not have an explicit graph structure. Their analysis suggests that under full supervision, kNN graphs do not offer benefits compared to a baseline that doe snot use the structure. The reviewers were mostly positive, nothing both theoretical and empirical contributions. While reviewers also had some concerns and questions, most of them were addressed during the rebuttal process. Overall, the consensus is that this is a valuable contribution  and should be accepted to the conference.